# Light intensity and opsin sensitivity shape the morphology of cone photoreceptor outer segments

Jingjin Xu[1,2,3], Zihan Chang[1,3], Wei Deng[1,3], Luwei Qian[1,3], Honggang Su[4], Xun Huang[4,5], Yunsi Kang[1,3], Haibo Xie[1,3], Chengtian Zhao[1,2,3]*

1 Fang Zongxi Center for Marine EvoDevo, MoE Key Laboratory of Marine Genetics and Breeding, College of Marine Life Sciences, Ocean University of China, Qingdao, China, 2 Laboratory for Marine Biology and Biotechnology, Qingdao National Laboratory for Marine Science and Technology, Qingdao, China, 3 Institute of Evolution & Marine Biodiversity, Ocean University of China, Qingdao, China, 4 State Key Laboratory of Molecular Developmental Biology, Institute of Genetics and Developmental Biology, Chinese Academy of Sciences, Beijing, China, 5 Tianjian Laboratory of Advanced Biomedical Sciences, Zhengzhou University, Zhengzhou, China

* chengtian_zhao@ouc.edu.cn

## Abstract

Regulation of neural cell morphology remains a fundamental question in neuroscience. Photoreceptor cells, a specialized class of neurons capable of initiating the phototransduction cascade, exhibit distinct structural and morphological characteristics. While the structural and morphological differences between rod and cone photoreceptors have been extensively studied, the variability in the morphology of cone outer segments (OS) remains largely unexplored. Zebrafish possess four distinct cone types, each displaying unique OS morphologies. By modulating opsin expression across cone types, we reveal that the morphology of the cone OS correlates directly with the wavelength sensitivity of the expressed opsins, with cones expressing longer wavelength-sensitive opsins exhibiting elongated OS. This regulatory mechanism is conserved across various vertebrates. Furthermore, we show that alterations in light intensity—induced by ectopic lipid droplet formation in the light path or by changing the environment light intensity—can also modulate OS morphology. Notably, this morphological plasticity is not transient, but rather dependent on long-term neural activity. Based on these findings, we propose a model for the regulation of cone OS length. Our data suggest that both opsin sensitivity and light intensity shape cone OS morphology through long-term neural activity, providing critical insights into neural plasticity in these light-sensitive neurons.

## Introduction

Neuronal plasticity, the ability of neuronal cells to alter their structure and shape, plays a fundamental role in brain development, learning, and memory. For example,

**Data availability statement:** All relevant data are within the paper and its Supporting information files.

**Funding:** This work was supported by the National Natural Science Foundation of China (https://www.nsfc.gov.cn/) (32125015 to C.Z., W2411026 to C.Z., 32500730 to J.X.) and funds from Laoshan Laboratory (LSKJ202203204 to C.Z). The funders had no role in study design, data collection and analysis, decision to publish, or preparation of the manuscript.

**Competing interests:** The authors have declared that no competing interests exist.

**Abbreviations:** dpf, days post-fertilization; LD, lipid droplet; LWS, long-wave sensitive opsin; mpf, month post-fertilization; OPL, outer plexiform layer; OS, outer segments; PFA, paraformaldehyde; ROS, rod outer segment; RPE, retinal pigment epithelium; TH, thyroid hormone; WGA, wheat germ agglutinin.

neurons can remodel their dendrites or axons in response to environmental stimuli or learning experiences. However, the mechanisms regulating these processes remain a central mystery in neurobiology [1–3]. Among specialized sensory neurons, photoreceptors are crucial for vision, converting light into electrical signals that serve as the foundation of visual perception. The OS of photoreceptors is critical for light detection as it contains photopigments that absorb light and initiate the visual process [4,5]. Based on OS morphology, photoreceptors are classified into two types: rods and cones. Rods enable vision in low-light conditions, while cones are responsible for color vision and high visual acuity in bright light. Despite their critical roles, the mechanisms underlying the formation and maintenance of such distinct OS morphology in rods and cones remain poorly understood.

Opsin proteins are the most abundant components of the OS in photoreceptor cells, where they bind to the chromophore retinal to initiate phototransduction. Notably, the OS is a dynamic structure that undergoes continuous and periodic renewal to sustain photoreceptor function [6,7]. Since the OS lacks the machinery for protein synthesis, its components must be synthesized in the cell body and transported to the OS through the connecting cilium—a narrow channel linking the inner and outer segments (OS). The OS can be considered a specialized form of cilium, as it shares several key features with ciliary structures in other cell types [8,9]. The transport of OS components is tightly regulated, and defects in this process are implicated in retinal diseases such as retinitis pigmentosa (RP) [10].

Downstream of opsin activation, rods and cones—despite their morphological differences—share a common phototransduction pathway. This involves the activation of transducin, cGMP phosphodiesterase, and the subsequent closure of cGMP-gated ion channels [4,11]. While rods often employ distinct homologous proteins for phototransduction, cones, irrespective of their subtype, largely utilize the same phototransduction components downstream of opsin activation [12]. The structural and morphological differences between rods and cones have been well-documented across various species [13]. However, the morphological diversity among the OS structures of cone subtypes and the regulatory mechanisms driving these differences remain largely unexplored.

Cone photoreceptors are indispensable for human vision, enabling us to perceive color and adapt to varying light conditions. Humans possess three cone types: S-cones (sensitive to short wavelengths, or blue light), M-cones (sensitive to medium wavelengths, or green light), and L-cones (sensitive to long wavelengths, or red light). In contrast, mice—being nocturnal—have a visual system optimized for low-light conditions and possess only two cone types: S-cones (sensitive to ultraviolet light, UV) and M-cones (green). These cones constitute only about 3% of the total photoreceptors, and their even distribution across the retina makes studying cone function in mice more challenging than studying rods [14]. By comparison, the zebrafish retina is predominantly cone-based, containing approximately 92% cones at the larval stage and about 60% in adults—closely resembling the cone-rich human macula [15–18]. Zebrafish possess four types of cone photoreceptors for color vision, which includes short-wavelength-sensitive (blue) cones, mid-wavelength-sensitive

(green) cones, long-wavelength-sensitive (red) cones, and ultraviolet-sensitive (UV) cones [15,19]. These cone types form a highly ordered mosaic in the adult zebrafish retina, with UV and blue cones alternating in rows, and green and red cones alternating in neighboring rows [20,21]. Although zebrafish and human cone subtypes are not directly evolutionarily equivalent—for example, human "green" and "red" cones both express LWS opsins and correspond to zebrafish red cones, whereas human "blue" cones (SWS1) are orthologous to zebrafish UV cones [22,23]—zebrafish nevertheless provide an excellent model for investigating cone photoreceptor morphology and development.

In this study, we used zebrafish as a model organism to explore the mechanisms that govern cone OS morphology. We show that there are significant length differences in the OS among various cone subtypes. By switching the expression of cone opsins between different photoreceptor subtypes, we demonstrate that cone OS morphology is directly regulated by the photosensitivity of the opsin protein. Moreover, changing the light intensity reaching the OS either genetically or environmentally can also modify the morphology of the cone OS. This regulatory mechanism appears to be conserved across multiple species. Our data suggest that cone OS morphology is finely tuned to adapt to light conditions, which underscores the dynamic interplay between structure and function in sensory systems.

## Results

### Variation in the morphology of cone OS in zebrafish

Similar to humans, the diurnal zebrafish primarily relies on cone photoreceptors for daytime vision. The zebrafish retina contains five types of photoreceptors: one type of rod and four types of cone photoreceptors (Fig 1A). Using wheat germ agglutinin (WGA) staining, we were able to distinguish the OS of the different photoreceptor types, which are organized into distinct layers (Fig 1B). In the apical region near the retinal pigment epithelium (RPE), the rod outer segment (ROS) layer is located at the topmost position, followed by the double- and blue-cone OS layers, with the UV cone OS layer positioned at the most basal location (Fig 1B and 1C).

To further investigate the morphology of the cone OS, we utilized two transgenic zebrafish lines, *Tg(sws1:sws1-GFP)* and *Tg(sws2:HA-tdTomato-CT44)*, to label the OS of UV and blue cones (Fig 1D and 1E). The OS of double cones (red and green cones) were identified using a combination of WGA and Zpr3 antibody staining. The Zpr3 antibody labels not only rod OS but also the OS of green and red cones [24], whereas WGA selectively labels red cone OS with strong fluorescence (Fig 1F). Measurement of the OS lengths revealed that rods had the longest OS, with an average length of approximately 35 μm (±3.76) (Fig 1C and 1G). Among the cone OS, the UV cone OS was the shortest, measuring approximately 10 μm (±0.49), while the blue cone OS had an average length of 12.5 μm (±1.33). Both red and green cone OSs were significantly longer than those of the blue and UV cones, with red cone OSs being slightly longer than green cone OSs (18.6 μm ± 0.94 versus 16 μm ± 1.13) (Fig 1G).

Additionally, we measured the widths of the different OSs, and found that the width of the cone OS was negatively correlated with its length (Fig 1H). A comparison of the ratio of OS length to width further highlighted significant morphological differences between the OSs of different photoreceptor types (Fig 1I). To further validate these observations, we performed ultrastructural analysis on adult zebrafish, which confirmed the morphological differences (S1 Fig). These results suggest that cone OS morphology in the zebrafish retina exhibits notable variation, both in terms of length and width.

### UV opsins are essential for the maintenance of UV cones

To investigate the mechanisms regulating OS length and considering that opsins are the most abundant proteins in the OS, we generated zebrafish mutants lacking cone opsins. We designed sgRNAs targeting the second exon of the UV opsin (*opn1sw1*, or *sws1*) gene, resulting in a frameshift mutation that led to the loss of the C-terminal 214 amino acids (Fig 2A). This mutation caused a significant reduction in mutant mRNA expression in 5 days post-fertilization (dpf) larvae, likely due to nonsense-mediated decay (Fig 2B) [25,26]. Immunostaining with WGA revealed that UV cone OSs failed to

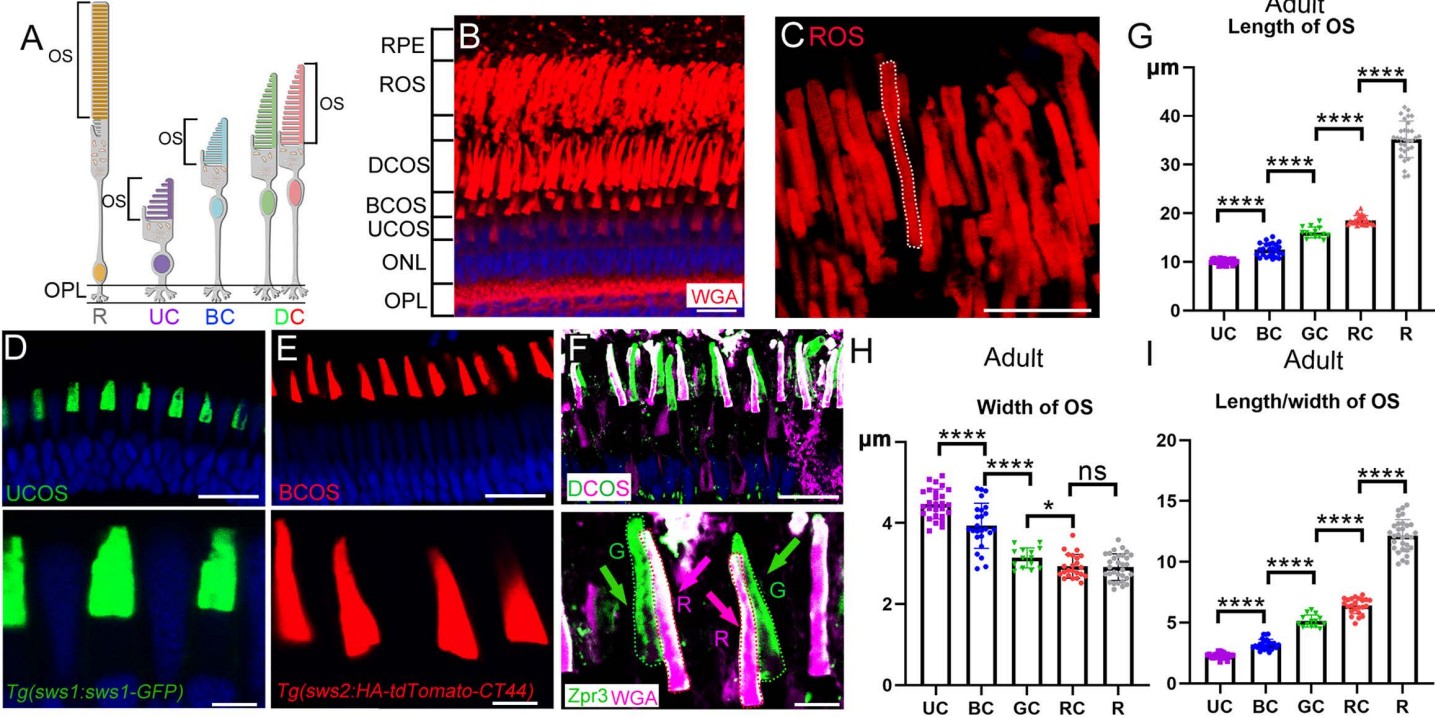

**Fig 1. Morphological diversity of cone OS in the zebrafish retina. (A)** Schematic diagram of rod and cone photoreceptors in the zebrafish retina. **(B)** Confocal image showing the distribution and morphology of photoreceptor OS, labeled with Alexa Fluor 555-conjugated wheat germ agglutinin (WGA). **(C)** High-magnification view of rod OS labeled with WGA. **(D)** Confocal image of UV cone OS labeled with *Tg(sws1:sws1-GFP)*. **(E)** Confocal image illustrating the morphology of blue cone OS, labeled with *Tg(sws2:HA-tdTomato-CT44)*. **(F)** Double cone OS (green/red) labeled by WGA and the Zpr3 antibody. Zpr3 labels double cone OS (green channel), while WGA labels the red cone OS with strong fluorescence (magenta channel). High-magnification views for panels **(D–F)** are shown below each respective image. **(G–I)** Quantitative analysis of OS dimensions, including length, width, and length-to-width ratio. In all panels, DAPI (blue) was used to stain cell nuclei. Abbreviations: R, rod; UC, UV cone; BC, blue cone; DC, double cone; RPE, retinal pigment epithelium; ROS, rod outer segment; DCOS, double cone outer segment; BCOS, blue cone outer segment; UCOS, UV cone outer segment; ONL, outer nuclear layer; OPL, outer plexiform layer. Scale bars, 20 μm in **(B–F)** and 5 μm for enlarged views. Data information: In (G–I), each dot represents one photoreceptor OS. Sample sizes per group are as follows: OS $n(UC) = 27$, $n(BC) = 25$, $n(GC) = 14$, $n(RC) = 23$, $n(R) = 31$. Data were derived from $N = 6$ zebrafish per group. Statistical significance was determined by one-way ANOVA with Bonferroni's post hoc test. * $p < 0.05$; **** $p < 0.0001$; ns, not significant. The data underlying this Figure can be found in S1 Data.

develop in the mutants at both larval and adult stages (Fig 2C and 2D). In the absence of OSs, UV cones began to degenerate around 7 dpf, as indicated by the loss of GFP expression driven by the *sws1* promoter. In adult fish, UV cones were nearly completely absent (S2A–S2E Fig).

## Ectopic expression of cone opsins in UV cones

To examine whether UV opsins can be functionally replaced by other opsin proteins, we first generated a GFP-tagged UV opsin transgene, *Tg(sws1:sws1-GFP)*. Although the GFP fusion protein was successfully targeted to the UV-cone OS, it failed to rescue UV-cone degeneration in the absence of Sws1 (S2F–S2I Fig). In contrast, UV-cone degeneration was fully rescued by the *Tg(sws1:sws1)* line expressing the full-length, untagged UV opsin (S2J and S2K Fig). These results suggest that the GFP tag may interfere with the structural integrity or proper function of UV opsin required for OS formation and maintenance.

Next, we asked whether ectopic expression of other opsins could rescue UV cone degeneration in the mutants. We generated zebrafish transgenes expressing green (*mws3*), red (*lws1*), and rod opsin genes. Strikingly, all these

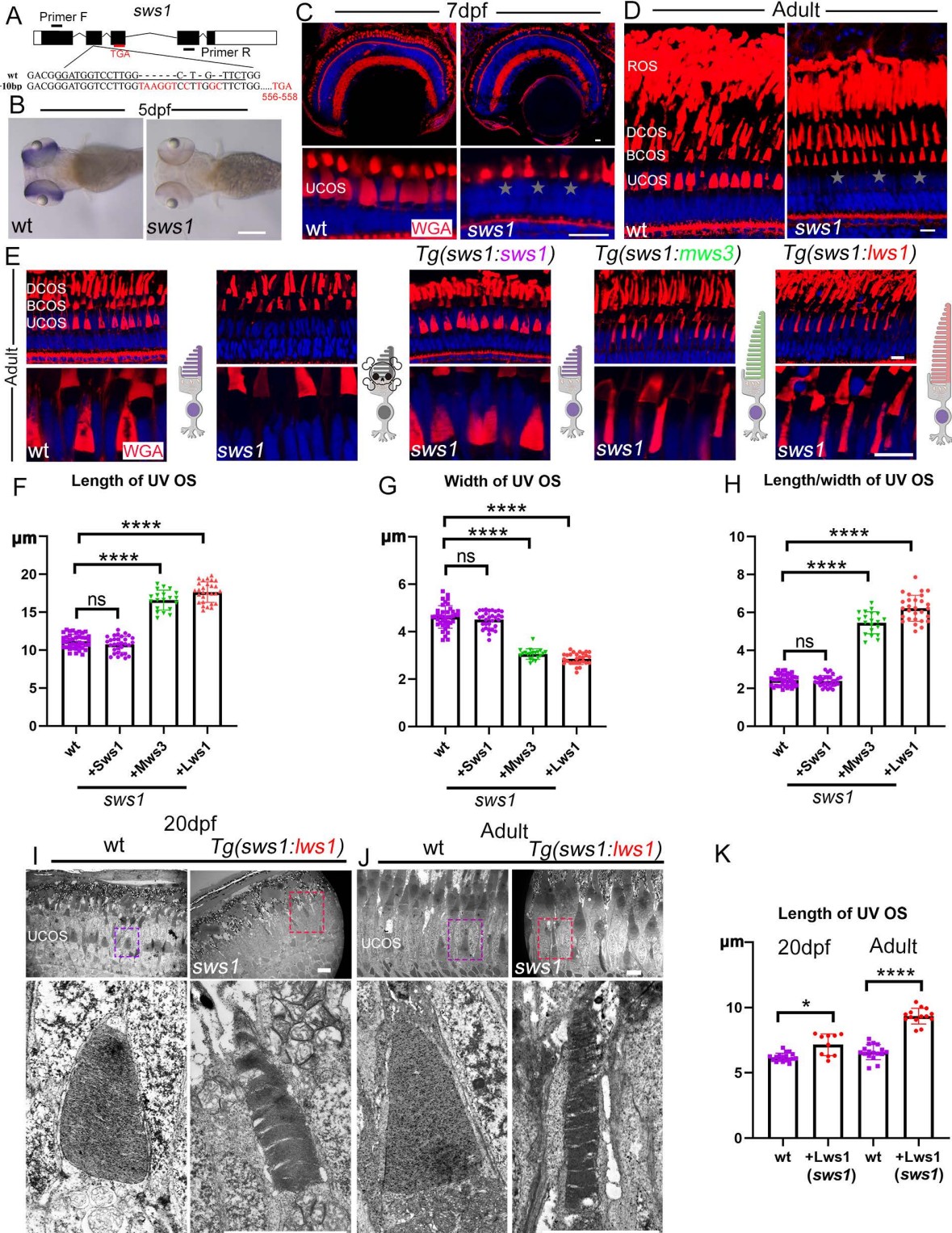

**Fig 2. Phenotypic analysis of *sws1* mutants and ectopic cone opsin expression in UV cones. (A)** Genomic structure of the zebrafish *sws1* gene, with wild-type and *sws1* mutant allele sequences shown below. The premature stop codon due to the frameshift mutation is indicated. **(B)** Whole-mount in situ hybridization showing *sws1* expression in 5 dpf wild-type and *sws1* mutant larvae. The position of primers used for probe synthesis is indicated

in panel **(A)**. **(C, D)** Confocal images of photoreceptor OS morphology in 7 dpf **(C)** and adult **(D)** wild-type and *sws1* mutant retinae, visualized with WGA staining. The UV OS layer is absent in mutants (stars). **(E)** Ectopic expression of different cone opsins in the UV cones of adult *sws1* mutants. OS were visualized with WGA staining. Diagrams on the right illustrate the corresponding UV cone morphologies. Enlarged views are shown at the bottom. **(F–H)** Quantification of UV cone OS across different transgenic backgrounds, as indicated. **(I, J)** TEM analysis of UV cone OS morphology in wild-type and *sws1* mutants carrying *Tg(sws1:lws1)* transgene at 20 dpf and adult stages. **(K)** Quantification of the length of UV cone OS. In panels **(C–E)**, DAPI (blue) marks cell nuclei. Scale bars: 200 μm in **(B)**, 10 μm in **(C–E)**, 5 μm in **(I, J)**. Data information: In **(F–H)**, each dot represents one photoreceptor OS. Sample sizes per group are as follows: OS $n$(wt) = 40, $n$(+Sws1) = 31, $n$(+Mws3) = 20, $n$(+Lws1) = 28. Data were derived from $N = 5$ zebrafish per group. Statistical significance was determined by one-way ANOVA with Bonferroni's post hoc test. In **(K)**, each dot represents one photoreceptor OS. Sample sizes per group are as follows: OS $n$(wt 20 dpf) = 18, $n$(+Lws1 20dfp) =10, $n$(wt adult) = 17, $n$(+Lws1 adult) = 14. Data were derived from $N = 3$ zebrafish per group. Statistical significance was determined by Kruskal–Wallis with Dunn's post hoc test. * $p < 0.05$; **** $p < 0.0001$; ns, not significant. The data underlying this Figure can be found in S1 Data.

transgenes rescued the loss of UV cone OS in the *sws1* mutants (Figs 2E and S3A). The number of UV cones was similar between the transgenes and wild-type siblings (S3B Fig). Remarkably, while the OS length is comparable between control and transgenic lines at larvae stages (S3C Fig), the rescued UV cone OSs exhibited significantly greater elongation in adult zebrafish upon ectopic expression of green or red opsin genes (Fig 2E and 2F). Statistical analysis showed that these rescued OSs resembled those of the green or red cone OSs of the double cones, although they localized to different layers of the retina (Fig 2F–2H versus Fig 1G–1I). Notably, ectopic expression of the rod opsin gene also rescued the cone OS, while the rescued OS resembles the morphology of the UV cone OS (S3A–S3D Fig). We further performed TEM ultrastructural analysis on *sws1* mutants carrying the *Tg(sws1:lws1)* transgene. The rescued "UV-cone" OS was correctly positioned beneath the blue-cone OS; however, its morphology was dramatically altered and closely resembled that of a double cone (Fig 2I–2K). Altogether, these results suggest that ectopic expression of different cone opsins can modify the morphology of UV cone OSs.

### Ectopic expression of cone opsins can modify the morphology of rod OSs

To investigate whether ectopic expression of cone opsins can affect the morphology of rod OS, we also generated *rhodopsin* (*rho*) mutants and confirmed reduced expression of *rhodopsin* in the mutant larvae (S4A and S4B Fig). Consistent with previous findings, mutation of the *rho* gene led to rapid degeneration of rod photoreceptors (S4C and S4D Fig) [27,28]. We then created two transgenic lines, *Tg(xops:rhodopsin)* and *Tg(xops:lws1)*, which express *rhodopsin* or *lws1* genes under the *Xenopus* rhodopsin gene promoter [16]. Expression of *rhodopsin* fully rescued the OS loss phenotype in the *rho* mutants (S4E–S4G Fig). Remarkably, ectopic expression of the red opsin gene also partially rescued rod OS formation in both larvae and adults (S4E–S4G Fig). Interestingly, these rescued OSs became slender and elongated in adults, resembling red-cone OSs (S4G–S4J Fig). Furthermore, the OSs of red opsin-expressing rod cells were predominantly positioned within the cone-OS layer, as indicated by their reduced distance to the outer plexiform layer (OPL) (S4G and S4K Fig). These results suggest that the cylindrical shape of rod OSs can be modified by simply switching the rod opsin with a cone opsin.

### The morphology of cone OS is directly related to the absorption wavelength of corresponding opsins

Given that different opsins absorb light at distinct wavelengths, we hypothesized that the morphology of photoreceptor OSs may be influenced by the wavelength of light absorbed by the opsin. Interestingly, zebrafish UV cones possess the shortest OS, corresponding to the shortest wavelength of light—UV light. In contrast, red cones have the longest OSs, which align with the longest wavelength of light—red light (Fig 3A) [31,32]. This raises the question: would the OS of UV cone cells elongate if they expressed an opsin that absorbs light at a much longer wavelength? To address this, we generated a transgene expressing the long-wave cone opsin (LWS) from the turtle *Trachemys scripta elegans* [33]. The absorption wavelength of this opsin is significantly longer than that of zebrafish red opsin (620 nm versus 558 nm)

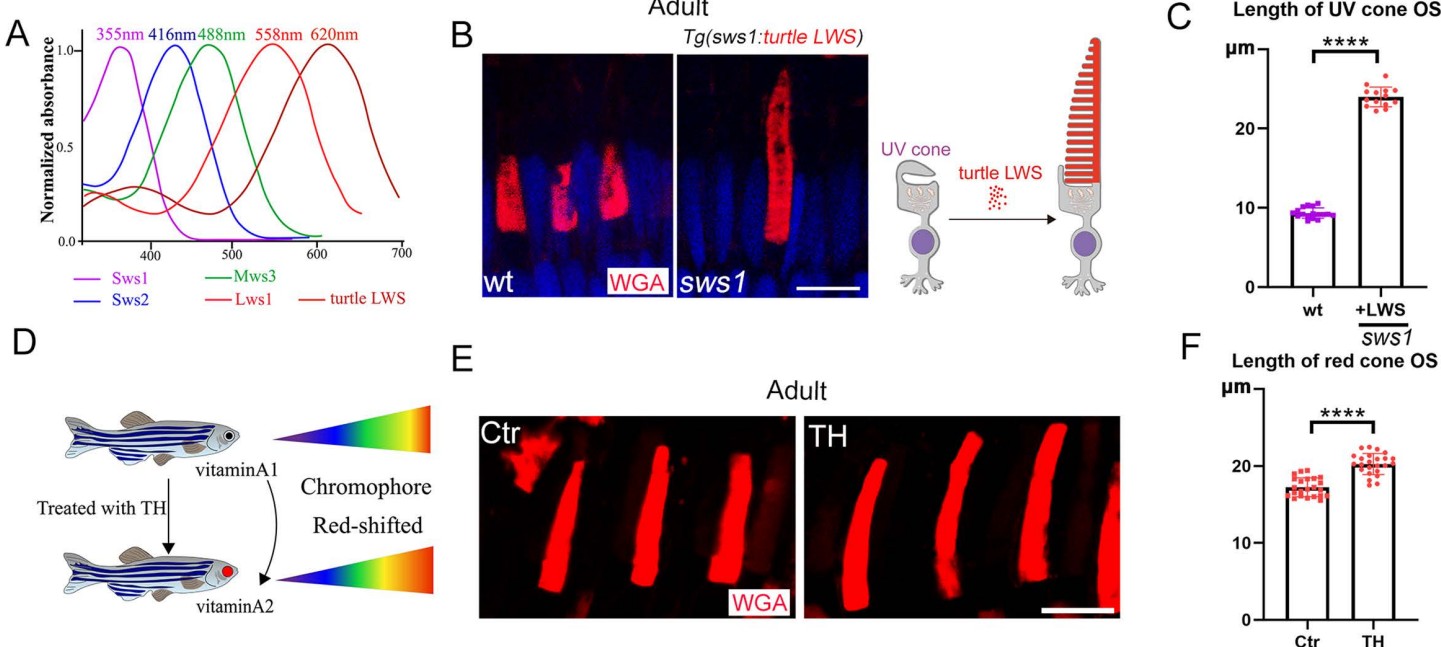

**Fig 3. Modification of cone OS morphology by the absorption wavelength of corresponding opsins. (A)** Absorption spectra of zebrafish cone opsins and the LWS opsin from *Trachemys scripta elegans* [29,30]. **(B)** Confocal images showing ectopic expression of turtle LWS opsin in the UV cones of *sws1* mutant zebrafish. UV cone OS were visualized with WGA staining. **(C)** Quantification of the length of UV cone OS. **(D)** Schematic diagram illustrating the red-shift in opsin absorption wavelength induced by thyroid hormone (TH) treatment. **(E)** Confocal images showing TH-induced morphological changes in red cone OS, labeled with WGA. **(F)** Quantification of the length of red cone OS after TH treatment. DAPI (blue) marks cell nuclei. Scale bars: 10 μm. Data information: In (C), each dot represents one photoreceptor OS. Sample sizes per group are as follows: OS $n$(wt) = 17, $n$(+LWS) = 14. Data for the wt and +LWS transgenic groups were derived from $N = 3$ and $N = 5$ zebrafish, respectively. Statistical significance was determined by the Mann–Whitney test. In **(K)**, each dot represents one photoreceptor OS. Sample sizes per group are as follows: OS $n$(Ctr) = 23, $n$(TH) =24. Data were derived from $N = 6$ zebrafish per group. Statistical significance was determined by the Student $t$ test. **** $p < 0.0001$. The data underlying this Figure can be found in S1 Data.

(Fig 3A). Remarkably, when turtle LWS was expressed in UV cone cells, these cells maintained their characteristic cone morphology but developed OSs 24 μm (±1.25) in length—substantially longer than the ~20 μm OS observed in zebrafish double-cone cells (Fig 3B and 3C).

Opsins require chromophores (retinal) to absorb light and undergo a conformational change that activates the phototransduction pathway. Notably, many aquatic animals, including fish, amphibians, and some reptiles, possess a Vitamin A1-A2 visual pigment system [34]. For instance, zebrafish photoreceptors predominantly contain Vitamin A1-based pigments, but treatment with thyroid hormone (TH) can induce a conversion to Vitamin A2-based pigments, resulting in the absorption of light at longer wavelengths (red-shifts) (Fig 3D) [35,36]. Strikingly, TH treatments also caused a pronounced elongation of red cone OS due to the red-shifts (Fig 3E and 3F). This further confirms that the length of photoreceptor OSs is linked to the wavelength of light absorbed by the opsin.

## Environmental adaptation of cone OS to different light intensities

According to optical theory, the wavelength of light is directly related to its energy, with shorter wavelengths, such as UV light, harboring the highest energy. The observed modification of cone OS morphology in response to changes in light wavelength suggests that light energy may play a critical role in determining OS length. To investigate further, we asked whether altering the intensity of light reaching the OS could similarly influence its length. To test this, we reared zebrafish

from one month post-fertilization (mpf) under either standard illumination (356 lx) or reduced illumination (38 lx), both maintained on a standard light-dark cycle (Fig 4A). Remarkably, after one month, zebrafish raised under dim light exhibited a significant elongation of cone OS compared to those reared under standard lighting conditions. This elongation was particularly pronounced in double cones (Fig 4B and 4C).

In various species, such as birds, cone photoreceptors have evolved a unique optical organelle known as the oil droplet or lipid droplet (LD), located within the inner segment. These oil droplets or LDs play a crucial role in photoreceptor activation by modifying the intensity of light transmitted to the OS [37]. These droplets consist of neutral lipids, primarily triacylglycerol, which act as intermediaries in the light path, thereby regulating the light intensity reaching the OS [37–39]. Consequently, we examined whether inducing the formation of oil droplets in zebrafish cone photoreceptors could modify the morphology of cone OS (Fig 4D). CIDEA (cell death-inducing DFFA-like effector A) is a critical factor in LD fusion and is highly expressed in avian cones [38,40,41]. CIDEA, together with SPDL1-L (spindle apparatus coiled-coil protein 1-L), a long isoform of SPDL1, regulates the formation and positioning of oil droplets in chicken cone photoreceptors [38] (Fig 4E). Differing from the well-studied SPDL1 short isoform (SPDL1-S), SPDL1-L contains a unique C-terminal region, which is essential for LD positioning [38]. Interestingly, the zebrafish *Spdl1* locus encodes only the *Spdl1-S* isoform (S5A Fig). Therefore, we generated a chimeric protein combining zebrafish Spdl1 with the C-terminal sequence of the human SPDL1-L protein (Figs 4F and S5A). We then generated a stable transgenic zebrafish line coexpressing the chimeric construct and zebrafish Cidea under the control of the *sws1* promoter, which is expected to induce LD formation specifically in UV cones (Fig 4F). At larval stages (10 dpf), LDs were observed in the inner segment of UV cones, and these LDs persisted in photoreceptors into adulthood (Figs 4G and S5B). Notably, some LDs began to degrade in certain UV cones during development, but a significant number of UV cones retained LDs (Figs 4G and S5B). Next, we compared the OS length of these photoreceptors. While no difference in OS length was observed between control and transgenic larvae at 10 dpf, a marked increase in OS length was noted in adult cones containing LDs (Figs 4H and S5C). Even within the same retina, UV cones containing LDs exhibited significantly longer OSs compared to those without LDs (Fig 4G and 4H). These findings suggest that ectopic LDs can also induce elongation of cone OSs. Although the composition of these LDs likely differs from that of native avian oil droplets, we speculate that they may still influence the light intensity reaching the OS, thereby promoting cone-OS elongation.

Finally, we asked whether such length adaption with light intensity occurs in natural conditions. In the ocean, light intensity decreases dramatically with the increase of depth due to absorption and scattering. The black rockfish (*Sebastes schlegelii*), a type of teleost found in the Northwest Pacific, can inhabit both deep and surface waters [42–44]. Wild-type black rockfish typically reside at depths ranging from 20 m to over 300 m [43,44]. In contrast, black rockfish is also cultivated as a farming fish in coastal regions, where they are raised in shallow waters (usually less than 10 m in depth). We compared cone OS between wild-type black rockfish, collected at depths of ~50 m, and cultured adult black rockfish raised in sea cages at ~7 m (Fig 4I). Using combined staining with WGA and the double cone-specific antibody Zpr1, we readily identified double cone OS in both wild-type and cultured fish (Fig 4J). Strikingly, the double cone OS in wild-type fish was also significantly longer than in cultured fish (Fig 4J and 4K). Altogether, these findings suggest that variations in light intensity also contribute to the regulation of cone OS length.

## Variation in cone OS morphology is conserved across species

We have demonstrated that the morphology of zebrafish cone OS varies significantly among different subtypes, with longer wavelength-sensitive cones having longer OS. Finally, we asked whether this pattern is unique to zebrafish or more broadly conserved. In teleosts, we observed similar length variations in cone OS length, including *Pterophyllum scalare*, *Cyprinus carpio*, and *Oryzias latipes* (S6A–S6F Fig). Remarkably, although cone width was comparable among these species, their length-to-width ratios among different cones differed substantially, potentially reflecting light adaptations to distinct ecological niches. We also examined cone OS morphology in rabbits, which possess two cone types sensitive

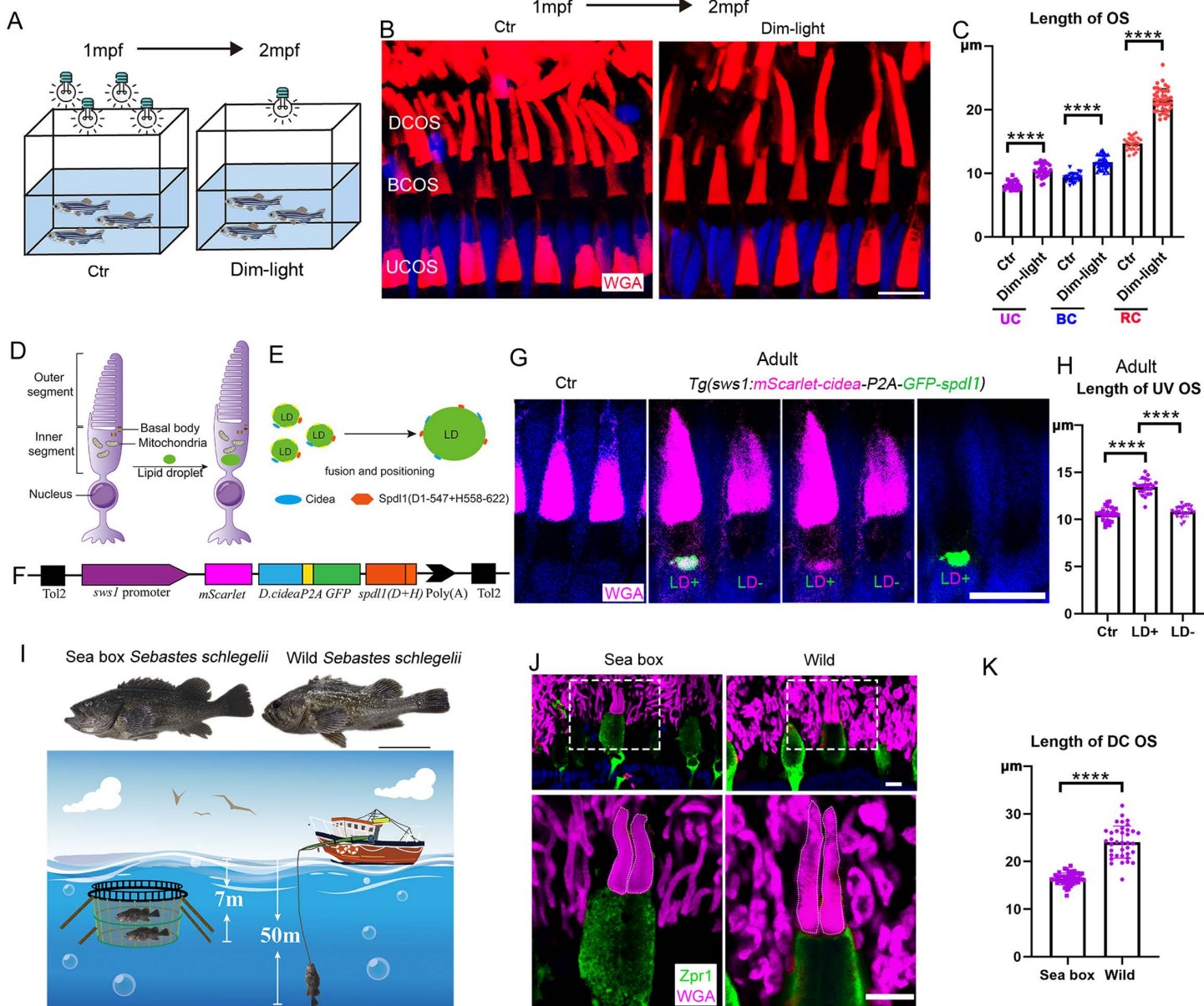

**Fig 4. Modification of cone OS morphology by light intensity. (A)** Schematic diagram of low light intensity treatments of zebrafish from 1 to 2 months post-fertilization. **(B)** Confocal images showing the lengths of cone OS in 2 mpf zebrafish following one month of exposure to normal and dim light. OS were labeled with WGA. **(C)** Quantitative analysis of cone OS lengths under different light intensities. **(D)** Schematic diagram illustrating ectopic lipid droplet expression in cone photoreceptor cells to modulate light intensity reaching the OS. **(E)** Diagram showing lipid droplet fusion mediated by Cidea and Spdl1 proteins. **(F)** Transgene construct for ectopic lipid droplet expression in UV cones: *Tg(sws1:mScarlet-cidea-P2A-GFP-spdl1)*. **(G)** Confocal images of ectopic lipid droplet expression at the inner segments of UV cones in adult zebrafish. UV cone OS were labeled with WGA (magenta), lipid droplets labeled with *Tg(sws1:mScarlet-cidea-P2A-GFP-spdl1)* (magenta and green). **(H)** Quantification of UV cone OS length in adult zebrafish. **(I)** Schematic diagram and live images showing the different living depths of aquaculture-reared (sea box) and wild-caught *Sebastes schlegelii*. **(J)** Confocal images comparing double cone OS morphology between aquaculture-reared and wild-caught *Sebastes schlegelii*. OS were labeled with WGA (magenta), and double cone cell bodies were marked with Zpr1 (green). **(K)** Quantitative analysis of double cone OS length in aquaculture-reared and wild-caught *Sebastes schlegelii*. DAPI (blue) marks cell nuclei. Scale bars: 10 cm in **(I)**; 10 μm in **(B, G, J)**. Data information: In **(C)**, each dot represents one photoreceptor OS. Sample sizes per group are as follows: OS $n$(Ctr UC) = 28, $n$(Dim-light UC) = 34, $n$(Ctr BC) = 24, $n$(Dim-light BC) = 39, $n$(Ctr RC) = 26, $n$(Dim-light RC) = 32. Data were derived from N = 5 zebrafish per group. Statistical significance was determined by one-way ANOVA with Bonferroni's post hoc test. In **(H)**, each dot represents one photoreceptor OS. Sample sizes per group are as follows: OS $n$(Ctr) = 27, $n$(LD+) = 24, $n$(LD−) = 23. Data for the wt and LD+ transgenic groups were derived from $N$ = 4 and $N$ = 6 zebrafish, respectively. Statistical significance was determined by

one-way ANOVA with Bonferroni's post hoc test. In **(K)**, each dot represents one photoreceptor OS. Sample sizes per group are as follows: OS *n*(Sea box DC) = 22, *n*(Wild DC) = 23. Data were derived from *N* = 3 *Sebastes schlegelii* per group. Statistical significance was determined by Student *t* test. **** *p* < 0.0001. The data underlying this Figure can be found in S1 Data.

to blue and green light [45]. Notably, these two cone types exhibited significantly different OS lengths in the rabbit retina (S6G–S6H Fig). Likewise, in the human fovea, S -cones also exhibit shorter OS lengths compared to M/L-cones, as reported in previous studies [46–48]. Together, these observations suggest that cone OS length variation is a conserved feature across a wide range of vertebrate species, from teleosts to mammals.

## Discussion

The morphology of photoreceptor OS represents a major structural distinction between rods and cones. Although these differences have been well described, the mechanisms that generate and maintain their distinct OS architectures remain poorly understood. Even less is known about the morphological variations between different cone OS subtypes. One of the primary challenges in addressing this question is the lack of suitable model systems for experimental investigation. As a nocturnal animal, mice possess a limited number of cones, and the co-expression of different cone opsins within the same cones further complicates the analysis of cone OS formation mechanisms [14]. In contrast, the zebrafish retina contains a large number of cones with four distinct subtypes, making it an excellent model for studying cone development.

Here, we show that zebrafish cone OS morphology can be modified simply by altering the photosensitivity of the opsin proteins. Notably, OS elongation does not appear to result from increased opsin accumulation. In *sws1* heterozygous mutants, UV cone OSs were comparable in size to those of wild-type adults, despite a substantial reduction in *sws1* expression (S7A–S7C Fig). Because specific antibodies against Sws1 are unavailable, we further quantified Sws1 protein levels using data-independent acquisition (DIA) mass spectrometry, which confirmed the decreased protein abundance in heterozygotes (S7D Fig). Moreover, the expression level of ectopic *lws1* (which produces a longer OS) was even lower than that of *sws1* in the *Tg(*sws1:sws1*)* line (which produces a shorter OS) (S7E Fig). Together, these findings indicate that factors beyond total opsin amount contribute to the regulation of cone OS morphology.

We propose several possible mechanisms that may influence OS length. First, subtle structural differences among opsins may affect their folding or interactions within the membrane, thereby altering disc composition or packing within the OS. Second, opsins are trafficked to the OS through a ciliary transport system, and transport efficiency may differ between cone subtypes. Such differences could also influence the recruitment of membrane-trafficking regulators, such as Rab or Arl13b proteins. Finally, the rate of photoreceptor disc shedding by RPE cells may vary across cone types, potentially contributing to differences in OS stability and length.

While all of these factors may contribute, we think the key regulator of OS modification is neural activity. This is supported by the finding that, at larval stages, UV-cone OSs in both the ectopic opsin-expression and LD induction experiments remain similar in length to those of wild-type controls (S8A, S8B, S3C, and S5C Figs). Notably, when *Tg(sws1:lws1)* fish were reared in complete darkness—conditions under which phototransduction-driven neural activity is minimized—the elongation of the "UV-cone" OS was significantly reduced (S8A and S8B Fig). Furthermore, even in adult fish, dim-light incubation increased cone-OS length (S8C and S8D Fig). These results highlight neural activity as a critical factor governing cone-OS morphological plasticity.

How might neural activity influence cone OS morphology? To transduce visual signals from photoreceptors to downstream bipolar cells, photoreceptors must continuously regulate the release of neurotransmitters such as glutamate at their synaptic terminals. This sustained synaptic output depends on adequate electrical responses generated by the phototransduction cascade. The first step in this cascade is the absorption of photon energy by opsin proteins in the OS (S9A Fig). Consequently, the OS must capture enough light energy to reach the threshold for signal initiation. Photon

capture efficiency depends on two main factors: the energy of individual photons and the number of photons captured by photopigments. Short-wavelength photons, which carry higher energy, are therefore likely to activate their corresponding opsins more efficiently. In contrast, switching opsin expression from a short-wavelength to a long-wavelength variant—for example, ectopic expression of *Lws1* in UV cones—may reduce activation efficiency during phototransduction. Similarly, a decrease in the number of photons reaching the OS, such as through the presence of additional LDs or under dim-light conditions, would be expected to further diminish opsin activation. To compensate for reduced activation efficiency, animals may adopt an alternative strategy—increasing the amount of opsin available in the light path, thereby enhancing photon capture within the OS. Such a mechanism could drive OS elongation (see detailed discussion in the legend of S9 Fig). Notably, this morphological adaptation likely requires prolonged adjustments and may be mediated by neural activity–dependent mechanisms. Although the molecular and cellular pathways that couple neural activity to OS remodeling remain unclear, feedback signals from downstream bipolar cells could potentially contribute to the regulation of OS length, an idea that merits future investigation.

In summary, our findings demonstrate that the morphology of photoreceptor OS is directly linked to the type of opsin it expresses. The longer the wavelength sensitivity of the opsins, the longer the cone OS length. Such regulation may be achieved via long-term environmental adaptation or neural activity. Our results provide new insights into the development of photoreceptor OS and offer a novel understanding of the factors influencing OS structure.

## Methods

### Ethics statement

All animal procedures in this study were strictly conducted in accordance with the Guideline for Ethical Review of Animal Welfare (GB/T 35892-2018) of the People's Republic of China. All animal experiments were approved by the Animal Care Committee of Ocean University of China (Animal protocol number: OUC2012316).

### Animals

All zebrafish strains were maintained at 28°C under a 14-hour light/10-hour dark cycle. Embryos were reared at 28.5°C in E3 medium (5 mM NaCl, 0.39 mM $CaCl_2$, 0.17 mM KCl, 0.67 mM $MgSO_4$, 0.1% methylene blue) according to standard protocols. To generate zebrafish *sws1* and *rho* mutants, we utilized CRISPR/Cas9 technology. The sgRNA target sequences were 5′-GGGATGGTCCTTGGCTGTTC-3′ and 5′-TGTACACCTCCTTGCACGGC-3′. Cas9 mRNA and sgRNA were synthesized following established protocols [49,50], and were co-injected into zebrafish embryos at the one-cell stage. All information regarding the transgenic zebrafish lines is provided in S1 Table, and the primer sequences are listed in S2 Table. Unless otherwise stated, all analyses on adult zebrafish were conducted using 3- to 6-month-old individuals.

Teleost species, including *Pterophyllum scalare*, *Cyprinus carpio*, and medaka (*Oryzias latipes*), were purchased from a local market in Qingdao, China. Wild-type *Sebastes schlegelii* were collected by fishing in the East China Sea at a depth of approximately 50 m, while cultured black rockfish were obtained from a local market in Qingdao, China.

### Whole-mount in situ hybridization and histological analysis

Whole-mount in situ hybridization experiments were performed following a standard protocol [51]. The primers used to amplify *sws1* and *rho* genes are listed in S2 Table. For histological analysis, zebrafish larvae at 5 dpf were fixed overnight in 4% paraformaldehyde (PFA) at 4°C. After fixation, the larvae were washed in PBST (PBS with 0.1% Tween-20), dehydrated through a graded ethanol series (50%, 75%, 85%, and 95% ethanol, each for 15 min), and embedded in JB-4 embedding medium (Polysciences) as previously described [52]. Cryosections were prepared at a thickness of 6 μm using a Leica cryostat. The sections were examined under a Leica stereomicroscope, and images were captured with a Leica digital camera.

## Immunohistochemistry

Larval sections in the manuscript were prepared using the cryosectioning method. For the immunostaining of cryosections, zebrafish larvae were fixed in 4% PFA overnight at 4°C. The fixed larvae were washed twice in PBST for 5 min each, infiltrated with 30% sucrose in PBS overnight at 4°C, embedded in OCT (Leica), frozen, and sectioned at a thickness of 12 µm. The following primary antibodies and dilutions were used: mouse Zpr1 (1:500, Zebrafish International Resource Center), Zpr-3 (1:500, Zebrafish International Resource Center). Although WGA546 (1:100; Invitrogen) is commonly used as a marker for rod OS, we found that it labels both rod and cone OS in zebrafish. Notably, the labeling intensity and pattern varied among cone subtypes, potentially reflecting differences in glycoconjugate composition among photoreceptor classes.

Adult fish sections were prepared using vibratome sectioning. To minimize potential effects from retinomotor movements in teleosts, adult fish were euthanized in ice water during the daytime (between 12:00 and 17:00). Eyes were then enucleated and immediately transferred to L-15 medium (Sigma). Under a stereomicroscope (Motic), the outer cornea and lens were carefully removed using forceps. The remaining retinas were fixed in 4% PFA at 4°C overnight. Retinas were then embedded in 4% low-melting-point agarose, sectioned into 40 µm-thick vibratome slices, and subjected to immunolabeling.

Upon capture, eyes from both wild and cultured black rockfish were immediately enucleated and transferred into L-15 medium (Sigma). The outer cornea and lens were carefully removed using forceps. The remaining retinas were fixed in 4% PFA at 4°C overnight. All subsequent steps, including vibratome sectioning and immunofluorescence staining, were performed using the same protocols as described for zebrafish retinas.

## Measurement of photoreceptor OS morphology

Photoreceptor OS were measured only from cone cells with a clearly identifiable OS tip. OS width was measured at the base of the OS, while OS length was measured from the base to the apex. For closely apposed double cones, particular care was taken to accurately define the OS tip to avoid errors caused by signal overlap. To ensure consistency, the OS tip was determined based on the following criteria: (a) The terminal point of WGA lectin labeling, identified as the distal-most location where a continuous and sharply demarcated WGA signal from an individual cone terminated. (b) For cases with significant lateral overlap where tips were indistinguishable in two-dimensional projections, the three-dimensional information from the Z-stack was utilized to trace each OS to its individual terminus.

## Transmission electron microscope

Retinae from adult zebrafish were fixed overnight at 4°C in 2.5% glutaraldehyde prepared in 0.1 M PBS. The samples were washed 3 times with PBS for 15 min each, followed by fixation in 1% osmium tetroxide for 2 hours. After fixation, the samples were washed, gradually dehydrated through increasing concentrations of acetone up to 100%, and embedded in Epon812 resin. Ultra-thin sections (70 nm) were cut using a Leica EM UC7 ultramicrotome. The sections were collected and stained with uranyl acetate and lead citrate. Specimens were then examined using a transmission electron microscope (Hitachi HT7700) and imaged with Olympus Soft Imaging Solutions.

## Quantitative PCR

Retinal tissues were collected from adult zebrafish. For each biological replicate, two adult zebrafish per group were euthanized by immersion in ice-cold water (<4°C). Four retinas were then dissected, pooled, and immediately processed for RNA extraction using Trizol reagent (Takara). RNA was reverse-transcribed using the HiScript III RT SuperMix for qPCR kit (Vazyme, #R323-01). qPCR was performed on a StepOne Real-Time PCR System (Thermo Fisher Scientific) with EvaGreen Master Mix (ABM). We estimated expression levels using the relative standard curve method, using five serial

standard dilutions of cDNA obtained from wild-type adult. Reactions were run in technical triplicate under the following cycling conditions: 95°C for 15 s, then 40 cycles of 95°C for 5 s, 60°C for 15 s, and 72°C for 35 s. Relative gene express levels were determined by the comparative Ct ($2^{-\Delta\Delta Ct}$) method, with zebrafish *β-actin* serving as the endogenous control. Primer sequences are listed in S2 Table. Differences between the two groups were assessed using the Mann–Whitney *U* test, a *p*-value <0.05 was considered statistically significant.

## Data-independent acquisition

To compare the protein expressiong level between wild-type and *sws1* heterzygotes groups, we performed Data-independent acquisition mass spectrometry. Briefly, four complete retinas were independently collected and pooled for each samples. Immediately after dissection, retinas were placed in pre-chilled 1.5 mL microcentrifuge tubes, flash-frozen in liquid nitrogen, and stored at −80°C until further processing. Protein extraction, tryptic digestion, and subsequent liquid chromatography–tandem mass spectrometry (LC–MS/MS) analysis were performed by Applied Protein Technology (Shanghai, China). The raw DIA data files were processed using DIA-NN software (version 1.8.1). Precursor and protein quantification was performed using the built-in quantification algorithms of DIA-NN. The quantified intensity values for the target protein, UV Opsin (Sws1, encoded by the *opn1sw1* gene), were extracted from the report for subsequent statistical analysis between the wild-type and *sws1*$^{+/-}$ groups. The raw mass spectrometry data are provided in Supporting Information S1 Data.

## TH treatments

The procedures for TH treatments were similar to those previously reported [36]. Briefly, L-thyroxine (Selleck) was dissolved in 0.1 M NaOH to prepare a stock solution at a concentration of $3 \times 10^5$ µg/L. For treatments of zebrafish, the L-thyroxine stock solution was diluted with fresh rearing system water to final concentrations of 300 µg/L. Zebrafish were maintained under a normal light cycle and feeding schedule, and the system water containing the drug was replaced every 24 hours for a duration of 2–3 weeks.

## Lower light exposure treatments

One- or 5-month-old zebrafish were maintained in light-isolated chambers containing 500 mL of system water at 28°C. The control group was exposed to standard-intensity illumination (356 lx) via LED panels (400–830 nm), whereas the low-intensity group was exposed to dim light (38 lx) using calibrated LED light sources. Illumination was provided on a 14 h light/10 h dark cycle, with lights on at 08:30 and off at 22:30. Zebrafish were fed twice per day with freshly hatched *Artemia nauplii* (brine shrimp). After 30 days of photoperiod conditioning, retinal sections were prepared for immunofluorescence microscopy to assess cone photoreceptor OS morphology.

## Dark treatments

Two-week-old wild-type and *Tg(sws1:lws1)* zebrafish were maintained in light-isolated chambers containing 500 mL of system water at 28°C. The control group was exposed to standard-intensity illumination (356 lx) via LED panels (400–830 nm). Illuminated groups were subjected to a 14 h light/10 h dark cycle, with lights on at 08:30 and off at 22:30. A dark group was kept in constant darkness throughout the experiment. Zebrafish were fed twice daily with freshly hatched *Artemia nauplii*. After 45 days of photoperiod conditioning, retinal sections were prepared for immunofluorescence microscopy to assess cone photoreceptor OS morphology.

## Statistical analysis

Data in all graphs are presented as mean ± standard deviation (SD). Normality was assessed for all quantitative datasets, and the appropriate statistical test was selected accordingly: parametric tests (unpaired Student *t* test for two

groups, one-way ANOVA test followed by Bonferroni's correction for multiple comparisons.) were used when normality assumptions were met; otherwise, non-parametric alternatives (Mann–Whitney test for two groups, Kruskal–Wallis test followed by Dunn's correction for multiple comparisons.) were applied. Details on sample sizes ($n$), test values ,and significance levels are provided in the respective figure legends. A $p$-value $< 0.05$ was considered statistically significant. No randomization, blinding, or masking was employed in the animal studies. All experiments were repeated at least three times independently. The underlying data for all statistical analyses can be found in Supporting information S1 Data. Statistical analyses were performed using GraphPad Prism 9 (version 9.5; GraphPad Software, San Diego, CA, USA).

## Supporting information

**S1 Fig. Ultrastructural analysis of photoreceptor OS in adult zebrafish. (A1, A2)** Transmission electron micrograph (TEM) showing the distribution of cone OS in the adult zebrafish retina. **(B–D)** Higher magnification TEM images illustrating the ultrastructure of OS in UV cones (B), blue cones (C), and double cones (D). **(E–G)** Quantitative analyses of OS morphology. Abbreviations: RC, red cone; GC, green cone; BC, blue cone; UC, UV cone. Scale bars: 5 μm in (A); 2 μm in (B–D). Data information: In (E–G), each dot represents one photoreceptor OS. Sample sizes per group are as follows: OS $n$(UC) = 9, $n$(BC) = 12, $n$(GC) = 6, $n$(RC) = 6. Data were derived from $N = 3$ zebrafish per group. Statistical significance was determined by one-way ANOVA with Bonferroni's post hoc test. $p < 0.05$; ** $p < 0.01$; **** $p < 0.0001$; ns, not significant. The data underlying this Figure can be found in S1 Data.
(TIF)

**S2 Fig. Progressive degeneration of UV cones in *sws1* mutants. (A–D)** Confocal images showing the morphology of UV cones in wild-type and *sws1* mutant retinas at different developmental stages, as indicated. Enlarged views are shown at the bottom. UV cone cell bodies are labeled with *Tg(sws1:GFP)* (green). White asterisks indicate regions lacking UV cone cell bodies. In (D), the red rectangle outlines the remaining UV cones in the ciliary marginal zone (CMZ), and the white rectangle highlights a region shown in the enlarged view below. Adult=4 months. **(E)** Quantitative analysis of UV cone cell numbers across different developmental stages. **(F)** Confocal images showing the morphology of UV cones in wild-type (labeled with WGA, red) and *Tg(sws1:sws1-GFP)* (green). **(G)** Quantification of UV cone OS length in adult zebrafish. **(H)** Representative confocal images showing the morphology of UV cones following exogenous expression of Sws1-GFP in the *sws1* mutant background. **(I)** Quantitative analysis of the number of UV cone OS. **(J)** Confocal micrographs showing that expression of zebrafish Sws1 fully rescues UV cones in *sws1* mutants. **(K)** Quantitative analysis of the number of UV cones in adult zebrafish. DAPI (blue) marks cell nuclei. Scale bars: 25 μm (low-magnification images); 10 μm (high-magnification images). Data information: In panels (E, I, K), each dot represents the number of UV cones from the section of a larvae or adult fish. We only collect one data for each sample. Sample sizes per group (E) are as follows: $N$(wt 5 dpf) = 4, $N$(*sws1* 5 dpf) = 7, $N$(wt 7 dpf) = 6, $N$(*sws1* 7 dpf) = 5, $N$(wt 21 dpf) = 5, $N$(*sws1* 21 dpf) = 4, $N$(wt 4 month) = 3, $N$(*sws1* 4 month) = 3. Statistical significance was determined by Kruskal–Wallis with Dunn's post hoc test. In panel (I) Sample sizes per group are as follows: $N$(wt 7 dpf) = 17, $N$(Sws1-GFP 7 dpf) = 8. Statistical significance was determined by the Mann–Whitney test. In panel (K), sample sizes per group are as follows: $N$(wt) = 6, $N$(+Sws1) = 6. Statistical significance was determined by the Mann–Whitney test. Relative # of UV cones per A.U (arbitrary units) is calculated by the number of UV cones per arbituary length of confocal images (116.25 μm). In panel (G), each dot represents one photoreceptor OS. Sample sizes per group are as follows: OS $n$(wt) = 36, $n$(Sws1-GFP) = 32. Data were derived from $N = 3$ zebrafish per group. Statistical significance was determined by the Student $t$ test. **** $p < 0.0001$; ns, not significant. The data underlying this Figure can be found in S1 Data.
(TIF)

**S3 Fig. Rescue of UV cone OS deficiency in *sws1* mutants via different opsin proteins. (A)** Confocal images illustrating the morphology of UV cone OS in 5 dpf zebrafish with different transgenic backgrounds, as indicated. **(B, C)** Quantitative analysis of the number and length of UV cone OS in 5 dpf zebrafish larvae. **(D)** Confocal images showing the

morphology of UV cone OS in adult zebrafish following ectopic expression of rhodopsin. The OS were labeled with WGA. DAPI (blue) marks cell nuclei. Scale bars: 15 µm (low-magnification) and 5 µm (high-magnification) in (A) and (D). Data information: In panel (B), each dot represents the relative number of UV cone OS from the section of a larvae. We only collect one data for each sample. Relative # of UV cone OS per A.U (arbitrary units) is calculated by the number of UV cone OS per arbituary length of confocal images (38.75 µm). Sample sizes per group are as follows: $N$(wt) = 5, $N$(+Sws1) = 5, $N$(+Mws3) = 5, $N$(+Lws1) = 6, $N$(+Rho) = 5. In panel (C), each dot represents one photoreceptor OS. Sample sizes per group are as follows: OS $n$(wt) = 15, $n$(+Sws1) = 15, $n$(+Mws3) = 17, $n$(+Lws1) = 15, $n$(+Rho) = 15. Data were derived from $N$ = 3–6 independent biological replicates per group. The data underlying this Figure can be found in S1 Data. (TIF)

**S4 Fig. Phenotypic analysis of *rho* mutants and ectopic cone opsin expression in rod cells. (A)** Genomic structure of the zebrafish *rho* gene, with wild-type and *rho* mutant allele sequences shown below. The premature stop codon resulting from the frameshift mutation is indicated. **(B)** Whole-mount in situ hybridization illustrating *rho* expression in 5 dpf wild-type and *rho* mutant larvae. The position of primers used for probe synthesis is indicated in panel (A). **(C)** Confocal images showing the morphology of rod cell bodies (green) and OS (magenta) in the retinae of 5 dpf wild-type and *rho* mutant larvae. Rod cell bodies were labeled with the *Tg(xops:GFP)* transgene, while OS were labeled with the *Tg(xops:mCherry-CT44)* transgene. **(D)** Confocal images illustrating photoreceptor OS distribution and morphology in adult wild-type and *rho* mutant retinas. Rod cell bodies were labeled with *Tg(xops:GFP),* and OS were visualized with WGA staining. Rod OS are absent in *rho* mutants (stars). **(E)** Confocal images illustrating the morphology of rod OS in 7 dpf zebrafish with different transgenic backgrounds, as indicated. Rod OS were visualized with *Tg(xops:mCherry-CT44)* (red). **(F)** Quantitative analysis of the number of rod OS per section in 7 dpf zebrafish larvae. **(G)** Ectopic opsin expression in rod cells of adult *rho* mutants. Rod OS were visualized with *Tg(xops:mCherry-CT44)*. Compared to the wild-type control and *rho* mutants, rhodopsin expression rescued rod OS morphology, whereas ectopic red opsin expression can partially rescue rods and induced a cone-like OS morphology. Enlarged views of boxed regions are shown below. **(H–J)** Quantification of rod OS morphology under different genetic backgrounds. **(K)** Statistical analysis of the distance from the base of the rod OS to the OPL, as shown in panels (G). DAPI (blue) marks cell nuclei. Scale bars: 200 µm in (B); 20 µm in (C, D, E, G). Data information: In (F), each dot represents the relative number of rod OS from the section of a larvae. Sample sizes per group are as follows: $N$(wt) = 16, $N$(rho) = 20, $N$(+Rho) = 12, $N$(+Lws1) = 12. Statistical significance was determined by Kruskal–Wallis with Dunn's post hoc test. In (H–J), each dot represents one photoreceptor OS. Sample sizes per group are as follows: OS $n$(wt) = 15, $n$(+Rho) = 14, $n$(+Lws1) = 12. Data were derived from $N$ = 4 zebrafish per group. Statistical significance was determined by Kruskal–Wallis with Dunn's post hoc test. In (K), each dot represents one photoreceptor OS. Sample sizes per group are as follows: $n$(wt) = 14, $n$(+Rho) = 15, $n$(+Lws1) = 19. Data were derived from $N$ = 4 zebrafish per group. Statistical significance was determined by Kruskal–Wallis with Dunn's post hoc test.** $p < 0.01$; *** $p < 0.001$; **** $p < 0.0001$; ns, not significant. The data underlying this Figure can be found in S1 Data. (TIF)

**S5 Fig. Induction of lipid droplet formation in UV cones. (A)** Schematic representation of the domain structures of SPDL1 in zebrafish (D.Spdl1), chicken (C.SPDL1-L), and human (H.SPDL1-L). The chimeric protein used to induce lipid droplet formation contains the N-terminal of zebrafish Spdl1 (1–547) plus the transmembrane domain of human SPDL1 (558–622). Domain predictions were performed using InterPro. **(B)** Confocal images showing ectopic lipid droplet expression in UV cones of 10 dpf zebrafish larvae. The split green and magenta channels were shown on the right with arrows indicate ectopic LDs in the cell body of UV cones. **(C)** Quantification of UV cone OS length in 10 dpf larvae zebrafish. DAPI (blue) marks cell nuclei. Scale bar: 10 µm in (B). Data information: In (C), each dot represents one photoreceptor OS. Sample sizes per group are as follows: OS $n$(Ctr) = 16, $n$(LD+) = 16. Data for the wt and LD+ transgenic groups were

derived from $N = 5$ and $N = 8$ zebrafish, respectively. Statistical significance was determined by the Mann-Whitney test. ns, not significant. The data underlying this Figure can be found in S1 Data.
(TIF)

**S6 Fig. Length variation of cone OS in different species. (A)** Phylogenetic tree depicting the evolutionary relationships among *Pterophyllum scalare*, *Cyprinus carpio*, *Oryzias latipes*, *Sebastes schlegelii*, and *Danio rerio*. **(B)** Confocal images illustrating the morphology of cone OS in different species of teleost fish. The photoreceptor OS were labeled with WGA (red), and double cones were labeled with Zpr1 (green). Nuclei were stained with DAPI (blue). **(C–F)** Statistical analysis of cone OS lengths in various teleost fish species. **(G)** Confocal images illustrating the distribution and morphology of photoreceptor OS in adult rabbits. WGA (red) labels rod cell OS, while Zpr3 (green) can label the OS of the two types of cone cells. **(H)** Statistical analysis of cone cell OS lengths in rabbits. Scale bar: 5 μm in (B);10 μm in (H). Data information: In (C), each dot represents one photoreceptor OS. Sample sizes per group are as follows: OS $n$(UC) = 30, $n$(BC) = 26, $n$(DC) = 51. Data were derived from $N = 3$ *Pterophyllum scalare* per group. Statistical significance was determined by one-way ANOVA with Bonferroni's post hoc test. In (D), each dot represents one photoreceptor OS. Sample sizes per group are as follows: OS $n$(UC) = 9, $n$(BC) = 22, $n$(DC) = 30. Data were derived from $N = 3$ *Cyprinus carpio* per group. Statistical significance was determined by one-way ANOVA with Bonferroni's post hoc test. In (E), each dot represents one photoreceptor OS. Sample sizes per group are as follows: OS $n$(UC) = 11, $n$(BC) = 23, $n$(DC) = 27. Data were derived from $N = 3$ *Oryzias latipes* per group. Statistical significance was determined by one-way ANOVA with Bonferroni's post hoc test. In (F), each dot represents one photoreceptor OS. Sample sizes per group are as follows: OS $n$(BC) = 33, $n$(DC) = 23. Data were derived from $N = 3$ *Sebastes schlegelii* per group. Statistical significance was determined by the Student $t$ test. In (H), each dot represents one photoreceptor OS. Sample sizes per group are as follows: OS $n$(BC) = 12, $n$(GC) = 16. Data were derived from $N = 3$ rabbits per group. Statistical significance was determined by the Mann–Whitney test. **** $p < 0.0001$. The data underlying this Figure can be found in S1 Data.
(TIF)

**S7 Fig. Gene expression analysis of ectopic opsins. (A)** Confocal images of UV cone OS in wild-type and *sws1*⁺ᐟ⁻ heterozygous mutant zebrafish. Nuclei are stained with DAPI (blue). **(B)** Quantification of UV cone OS length in wild-type and *sws1*⁺ᐟ⁻ zebrafish. **(C)** qPCR of relative *sws1* mRNA expression levels in wild-type and *sws1*⁺ᐟ⁻ zebrafish. **(D)** DIA of relative *sws1* protein expression levels in wild-type and *sws1*⁺ᐟ⁻ zebrafish. **(E)** Relative mRNA expression of *sws1* and *lws1* in transgenic rescue lines. Scale bars: 15 μm (low-magnification) and 10 μm (high-magnification) in (A). Data information: In (B), each dot represents one photoreceptor OS. Sample sizes per group are as follows: OS $n$(wt) = 28, $n$(*sws1*⁺ᐟ⁻) = 28. Data were derived from $N = 5$ zebrafish per group. Statistical significance was determined by the Student $t$ test. In (C, E), the qPCR for each sample was performed in technical triplicates. ns, not significant. The data underlying this Figure can be found in S1 Data.
(TIF)

**S8 Fig. Elongation of the cone OS is dependent on neural activity. (A)** Confocal images showing the morphology of cone OS in wild-type or *sws1* mutants carrying *Tg(sws1:lws1)* transgene. The schematic diagram of the strategy of the light/dark treatment is shown on top of each figure. Enlarged views of the UV cone OS is shown on the bottom. OS were labeled with WGA. **(B)** Quantification of UV cone OS lengths under normal light and dark conditions with different genetic background as indicated. **(C)** Confocal images showing the lengths of cone OS following one month of exposure to normal and dim light starting from 5mpf old adult zebrafish. Schematic diagram of low-light intensity treatments was shown on the top. OS were labeled with WGA. To distinguish the length of blue and double cone OS, different focus planes were shown on the right. Z-stack image shows the maximum intensity projection image. **(D)** Quantitative analysis of cone OS lengths under different light intensities. DAPI (blue) marks cell nuclei. Scale bars: 10 μm in (A, C). Data information: In panel (B), each dot represents one photoreceptor OS. Sample sizes per group are as follows: OS $n$(wt 7dpf) = 26, $n$(+Lws1 7dpf)

= 30, $n$(wt 2mpf) = 40, $n$(+Lws1 2mpf light) = 39, $n$(+Lws1 2mpf Dark) = 31. Data were derived from $N$ = 6 zebrafish per group. Statistical significance was determined by one-way ANOVA with Bonferroni's post hoc test. In (D), each dot represents one photoreceptor OS. Sample sizes per group are as follows: OS $n$(Ctr UC) = 36, $n$(Dim-light UC) = 40, $n$(Ctr BC) = 29, $n$(Dim-light BC) = 32, $n$(Ctr RC) = 31, $n$(Dim-light RC) = 34. Data were derived from $N$ = 5 zebrafish per group. Statistical significance was determined by one-way ANOVA with Bonferroni's post hoc test. **** $p < 0.0001$; ns, not significant. The data underlying this Figure can be found in S1 Data.
(TIF)

**S9 Fig. Model for the regulation of cone OS length by Light intensity and opsin sensitivity. (A)** Schematic illustrating the path of light through the layered zebrafish retina. The different arrangement of rod and cone OSs in the photoreceptor layer were shown. **(B–E)** Schematic diagrams illustrating a model for how cone photoreceptor OS length is regulated. For effective phototransduction, the outer segment must generate sufficient membrane potential through cyclic nucleotide-gated (CNG) channels (right). Closure of these channels may require a minimum number of activated opsin molecules. For example, at least four activated opsins (yellow) may be needed to induce adequate CNG channel closure. Because short-wavelength ultraviolet (UV) light has higher photon energy, it can meet this activation threshold with only a single opsin layer (B). In contrast, long-wavelength red light, which activates red opsins less efficiently, requires additional opsin layers to capture enough photons to reach the threshold (C). Likewise, reduced light intensity—whether caused by lipid droplets or low-light environmental conditions—decreases the likelihood of photon capture, potentially necessitating additional opsin layers along the light path to maintain sufficient activation (D). Finally, in the absence of light, such elongation of the OS may not occur even when long-wavelength red opsins are expressed (E).
(TIF)

**S1 Table. Transgenic zebrafish generated in this work.**
(DOCX)

**S2 Table. Primer sequences.**
(DOCX)

**S1 Data. Raw data used in all figures.**
(XLSX)

## Acknowledgments

We would like to thank Dr. Jie Qi for her kind help during the acquirement of the black rockfish. We also thank Dr. Xungang Tan, members of the IEMB and FANG center for their kind help during the preparation of this manuscript. We also gratefully acknowledge the outstanding support from the core facilities of the IEMB and FANG Center at OUC.

## Author contributions

**Conceptualization:** Jingjin Xu, Xun Huang, Chengtian Zhao.

**Data curation:** Jingjin Xu.

**Formal analysis:** Jingjin Xu.

**Funding acquisition:** Jingjin Xu, Chengtian Zhao.

**Investigation:** Jingjin Xu.

**Methodology:** Jingjin Xu, Chengtian Zhao.

**Project administration:** Jingjin Xu, Chengtian Zhao.

**Resources:** Honggang Su, Xun Huang, Chengtian Zhao.

**Software:** Luwei Qian.

**Supervision:** Chengtian Zhao.

**Validation:** Jingjin Xu, Zihan Chang, Wei Deng.

**Visualization:** Jingjin Xu, Yunsi Kang, Haibo Xie.

**Writing – original draft:** Jingjin Xu, Chengtian Zhao.

**Writing – review & editing:** Chengtian Zhao.

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
