## [Editor Report · Decision Letter 0]

1 Jul 2025

Dear Dr Zhao,

Thank you for submitting your manuscript entitled "Light intensity and opsin sensitivity shape the morphology of cone photoreceptor outer segments" for consideration as a Short Report by PLOS Biology.

Your manuscript has now been evaluated by the PLOS Biology editorial staff, as well as by an academic editor with relevant expertise, and I am writing to let you know that we would like to send your submission out for external peer review.

However, we noticed that some of your supplemental figures are mentioned only in the Discussion section, when they should be brought up in the Results section. Moreover, we could not find a complete description of the methods used for all of these experiments and analyses.

It is essential that these issues are addressed before we send your paper out for review. For the next stage of submission, can you please ensure that all methods are described in detail in the Methods section, and that all results are discussed in the Results section? We have extended our normal deadline for the next stage of submission to 1 week to give you time to make these changes. Please let us know if you will require additional time, in which case we will withdraw your paper, and you can re-submit when you are ready (and we will then send it out for review without delay).

In addition, before we can send your manuscript to reviewers, we need you to complete your submission by providing the metadata that is required for full assessment. To this end, please login to Editorial Manager where you will find the paper in the 'Submissions Needing Revisions' folder on your homepage. Please click 'Revise Submission' from the Action Links and complete all additional questions in the submission questionnaire.

Once your full submission is complete, your paper will undergo a series of checks in preparation for peer review. After your manuscript has passed the checks it will be sent out for review. To provide the metadata for your submission, please Login to Editorial Manager (https://www.editorialmanager.com/pbiology) within two working days, i.e. by Jul 08 2025 11:59PM.

Kind regards,

Taylor

Taylor Hart, PhD,

Associate Editor

PLOS Biology

thart@plos.org

---

## [Decision Letter · Decision Letter 1]

14 Aug 2025

Dear Dr Zhao,

Thank you for your patience while your manuscript "Light intensity and opsin sensitivity shape the morphology of cone photoreceptor outer segments" was peer-reviewed at PLOS Biology. It has now been evaluated by the PLOS Biology editors, an Academic Editor with relevant expertise, and by several independent reviewers.

In light of the reviews, which you will find at the end of this email, we would like to invite you to revise the work to thoroughly address the reviewers' reports.

As you will see below, the reviewers praise the experimental design and say that the results are interesting. However, R1, R2 and R4 pointed out areas requiring additional support, including through adding control experiments, further quantifications, and addressing issues with the methodological descriptions, statistics, and figures. The reviewers all commented on the limited mechanistic insight, and several of them suggested further experiments that could provide further clarity here. The reviewers also offered suggestions for improving the framing of the study.

Given the extent of revision needed, we cannot make a decision about publication until we have seen the revised manuscript and your response to the reviewers' comments. Your revised manuscript is likely to be sent for further evaluation by all or a subset of the reviewers.

**IMPORTANT - SUBMITTING YOUR REVISION**

*Re-submission Checklist*

*Published Peer Review*

*PLOS Data Policy*

*Blot and Gel Data Policy*

Sincerely,

Christian

Christian Schnell, Ph.D.

Senior Editor

PLOS Biology

cschnell@plos.org

on behalf of

Taylor

Taylor Hart, PhD,

Associate Editor

PLOS Biology

thart@plos.org

REVIEWS:

Reviewer #1 (Juan Angueyra signed his report): SUMMARY OF REVIEW:

This paper aims to highlight the role of visual opsins in photoreceptor outer segment (OS) morphology, with the goal of providing insight to outer segment development and plasticity. Through a series of novel experiments in both larval and adult zebrafish, using new mutant and transgenic lines, the authors demonstrate that OS shape is correlated with opsin expression, opsin sensitivity, and light intensity. In addition, the authors carry additional manipulations (hormonal and environmental) and cross-species comparisons known to create differences in opsin expression and find corresponding changes in OS shape.

These results are exciting and interesting.

Finally, the authors propose a model on how photoreceptor OS is regulated by light energy. We believe that some changes in rephrasing this claim, and acknowledging the possibility of other mechanisms would improve the overall impact and readability, especially for non-retina experts. We also suggest the authors refine the figure legends and Methods to ensure the reproducibility.

What are the main claims of the paper and how significant are they for the discipline? Are these claims novel?

Claim 1: Specific opsin expression drives OS morphology.

Novel = Yes. Although the influence of opsin levels on outer segment length is well described, this study includes clever ectopic expression of opsins, which validates a mechanism where specific properties of each opsin control OS morphology

Claim 2: Correlation between OS length and wavelength sensitivity of opsin expressed

Novel = Yes. Cell-specific manipulations added novelty to well-described literature.

Claim 3: Control of OS shape is activity dependent

This mechanism is not well supported by the presented data.

Is this paper outstanding in its discipline?

Yes. This study uses novel and insightful approaches, presents interesting findings, and opens multiple possibilities for follow-up studies. In particular this will lead to experiments to identify mechanisms able to control cell shape, which is relevant to many fields.

Who would find this paper of interest? Why?

This could be of general interest to cell biologists, retinal biologists, and visual neuroscientists. Researchers interested in understanding the mechanisms that govern morphology differences across cell subtypes, particularly in the retina, will find this particularly interesting because it may provide a new bridge between structure and function in neurons.

Are the claims properly placed in the context of the previous literature? Have the authors treated the literature fairly?

Yes, the authors cite relevant literature to their field and everything is discussed in context to previous literature.

Do the data and analyses fully support the claims? If not, what other evidence is required?

Claim 1: Specific opsin expression drives OS morphology.

Supporting evidence: Yes; ectopic, environmental, and hormonal manipulations. In addition, cross-species comparisons. All lines of evidence reach similar conclusions.

Claim 2: Correlation between OS length and wavelength sensitivity of opsin expressed

Supporting evidence: Yes.

Claim 3: Control of OS shape is activity dependent

Supporting evidence: No, the evidence provided does not uniquely support this claim. We recommend the authors to modify the claim as it stands.

Further Evidence Required:

To support this claim, a manipulation that modifies activity is required. For example, mutations in phototransduction proteins that cause blindness could be a first step towards this goal.

Other possible (non-activity dependent) mechanisms need to be excluded. For example, OS morphology could be dictated by sequence differences in opsins, differences in ciliary transport of the misexpressed opsins, or changes rates of disc shedding by the RPE, amongst others.

Would additional work improve the paper? How much better would the paper be if this work were performed and how difficult would it be to do this work?

We recommend revisions of the text, without additional experiments.

If the paper is considered unsuitable for publication in its present form, does the study itself show sufficient potential that the authors should be encouraged to resubmit a revised version?

The authors should resubmit a revised version after thorough revision of the methods section to make the manuscript suitable for publications (see below). In addition, the authors should revise or soften their claim regarding that the mechanism that underlies OS length is activity dependent as the data is suggestive but insufficient to exclude other possible mechanisms.

Are original data deposited in appropriate repositories and accession/version numbers provided for genes, proteins, mutants, diseases, etc.?

N/A

Is the manuscript well organized and written clearly enough to be accessible to non-specialists?

Yes.

Are details of the methodology sufficient to allow the experiments to be reproduced?

No.

Imaging: not enough details provided. We recommend that the authors follow established standards in the field for reporting methods for fluorescent and electron-microscopy images (e.g. https://doi.org/10.1038/s41592-021-01156-w)

qPCR: key steps of protocol, experiments and analysis have been omitted. For example, no primers were provided for the normalization gene, no mention of RNA extraction protocol, of quality checks on cDNA, etc. (e.g. https://www.biorxiv.org/content/10.1101/2024.12.04.626769v1.full.pdf)

Light exposure experiments: no spectral data provided for LEDs used. Given the claims about spectrum and activity-dependent mechanisms, this information is highly relevant.

Statistics: none of the experiments have explicit mention of biological replicates, and relevant details to assess confidence on statistical analysis are omitted, including details (other than p-value) of statistical tests (e.g. https://pmc.ncbi.nlm.nih.gov/articles/PMC6639881/; https://plos.org/resource/how-to-report-statistics/)

Animals: While the results mention specific days post-fertilization, the manuscript lacks clarity regarding which developmental time points were used for comparative analysis with teleost species. Given that there is both larval and adult data presented, explicit mention of age is required for all datasets.

We recommend a thorough revision of the methods section. This should also include revision of figure legends; in particular "Quantitative analysis of …" should be replaced with all the details about statistical tests (n, degrees of freedom, value of t-test, etc.). There is also no description of what each dot in bar plots represents. If each dot represents a single OS, it is required to report how many images were analyzed for each experiment and if the images correspond to different animals or not (which correspond to true biological replicates), to be able to assess if this study has enough statistical power.

PLOS Biology encourages authors to publish detailed protocols and algorithms as supporting information online. Do any particular methods used in the manuscript warrant such treatment?

No.

Other Minor Edits:

Many images are not color-blind friendly. We recommend using magenta instead of red when red-green distinctions are important

Much of the text on the fluorescent images is difficult to discern. For example, the red WGA text can be difficult to read where it overlays the images, and would benefit from improved contrast or alternative labeling approaches.

Unclear if scale bar is consistent, and this is sometimes not mentioned. For example, in Figure 2 the dimensions for the scale bar is not mentioned for F-I even though there is a scale bar for I shown.

Having a key for colors in box plots could be helpful, and could be kept consistent for similar parameters in other plots.

Properly report the replicate size and the sample size.

Figure 2A; does the 10 bp insertion create a premature STOP codon?

Figure 3: Are both WGA and cidea labeled red? Is Cidea expressed in the transgenic cones?

Figure S3A: Why is there no quantification with GFP14? Why 5 dpf for only this experiment?

Figure S3G: Why is there no quantification? Is the rescue full? The example images seem to suggest that there are more UV cones in the rescue than in WT?

Figure S4B: What is relative #? How is this calculated and why is it not absolute?

Page 13: "Similar to humans, zebrafish cones are classified into S-cones (blue), M-cones (green), and L-cones (red) (Perkins & Fadool, 2010). Additionally, zebrafish feature a specialized type of cone that is sensitive to UV light" is misleading since the human S-cone is actually related to the zebrafish UV cone and not the Scone (see Baden et al, 2025: https://doi.org/10.1371/journal.pbio.3003157)

Include the spectrum of LED panel used for light exposure experiment

"Reducing photon energy (e.g., switching to longer wavelengths) or decreasing photon capture (e.g., via additional lipid droplets or in low-light environments) lowers the likelihood of photopigment activation": this is not true. For example, according to the spectral sensitivity curves for L-opsins, switching from blue light to red light (high energy to low energy) increases photopigment activation. Photopigment activation depends on the intensity AND spectrum of the stimulating light AND the spectral sensitivity of the opsin.

Suggestions:

Hebbian plasticity: The classic Hebb's rule is about the connection between neurons, and although famous, not the only example of activity dependence. We find it challenging to understand the direct comparison between Hebbian plasticity between two cells and the changes in cell shape presented in this manuscript.

Restructure the discussion to match flow of the introduction:The authors discuss environmental changes as a factor for altering OS shape. We recommend that the authors move the environmental changes section before discussion about oil droplets. The reviewers believe this structure will benefit the flow of the manuscript. If not, alter the introduction to highlight cellular modifications to improve/alter/enhance particular functions (such as oil droplets in photoreceptors).

Clarify times of experiments and adult vs larval data: The authors provide insightful data from both adult and larval stages of zebrafish, but it becomes unclear which pieces of data are being referred to. For example it is explained that the five photoreceptor subtypes are easily distinguished using wheat germ agglutinin (WGA) staining, but it is unclear at what developmental timepoint this is at. We suggest the authors add age to every figure once and distinguish when different ages are used. This will make the results explicitly clear when the authors are referring to morphology in adult vs. larval/developmental stages.

Suggestion for the discussion: the authors state that OS morphology (mainly length, width, and shape) is directly related to the opsin that is expressed. But does not acknowledge scenarios where opsins are co-expressed in the same cells (mice M/S gradient, cichlid/salmon UV/S transition). Do the authors think that OS morphology changes in hybridized cell identities?

Suggestion for the discussion: The term morphology encompasses quite a bit when looking at photoreceptor cell structure: in addition to the outer segment, did the authors notice any changes in the rest of the cell morphology? (inner segment, cell bodies, ribbon synapses).

Reviewer #2: Photoreceptors have outer segment (OS) regions specialized to absorb photons and create a visual response. The manuscript by Xu et al presents a set of data to support an interesting hypothesis, that cone photoreceptor OS size and shape is directly related to light sensitivity, and is adaptable depending on the conditions. Using zebrafish as a model, the authors characterize the OS dimensions of the four cone subtypes, demonstrating that photoreceptors detecting longer wavelengths have longer and wider OSs. The replacement of UV opsin in the short cones with another cone opsin was sufficient to convert the OS morphology to match the opsin. Surprisingly, rod opsin did not change the UV cone morphology. On the other hand, expression of red opsin in rods lacking rho created rods with OSs shaped similar to red cones and also shifted the OS position to a more inner layer. Expression of a long wavelength turtle opsin in sws1-deficient UV cones led to a tripling in OS length compared to wildtype UV cones. Treatment with thyroid hormone, which promotes the switch from Vitamin A1 to A2-based photopigments (A2 is red-shifted) promoted lengthening of red OSs. Another experiment involved the triggering of lipid droplet formation in the inner segment or growing fish in reduced light conditions, in both cases limiting light passage into the OS. The hypothesis was that the OS would lengthen to accommodate the reduction in light input, which was indeed the finding. Finally, the authors examined a variety of fish species, observing similar relationships between cone height and wavelength. For black rockfish that lives at depth when wild but is cultivated in shallower waters, the cone photoreceptors were longer in the deep-dwelling population. Overall, the authors used multiple experimental approaches to support their hypothesis that cone OSs adapt to their light-detecting functions.

Generally, the paper is well written and organized. Some of the experiments are quite ingenious. The group uses a wide breadth of experiments to support their hypothesis. My comments are mostly about improvements for the experimental analysis.

Major comments:

1) The transgenic line sws1:sws1-GFP is used to mark OS. But the line is massively overexpressing sws1 opsin in these cells. Evidence is needed to show that morphology is unaffected in these cells.

2) The WGA labelling is curious. WGA is typically used as a rod OS marker, so the specific labelling of cone OS is surprising. Can the authors explain or justify the discrepancy? Is there evidence in the literature for its use as a cone OS label, especially for UV cones.

3) What are the data points shown in the graphs throughout the paper? Are they measurements for individual cells, section, eyes, or fish?

4) t-tests are the only statistics described, which is not appropriate for multiple groups compared to one wildtype group, unless with a Bonferroni correction.

5) If the UV cones with swapped opsins remain in the same layer, then TEM could be used to support the conclusion of altered morphology and would be a strong addition, particularly for seeing changes in width of the UV cone OS.

6) The rho-/- retinas with lws1 expression must look vastly different from wt, and even the rho rescue looks disorganized. TEM would again benefit this characterization, and maybe also further immuno characterization of the retinal layering.

7) In Figure 2, Panel G, it looks like the top of the UV cone OSs are cut off, either by the physical or optical section. This is also true for the wt image in Fig 3, panel I and Fig 4B. How is the analysis controlled to ensure that the whole cell is captured? And if these images are correct, then are there changes to OS shape? This is an important point since the length of the OS is a key measurement in the analysis.

Minor comments:

1) There are no page or line numbers, which makes it difficult to provide clear comments.

2) Intro: The OS is not analogous to a dendrite. Receives signals, but very different structure.

3) Intro: The authors state the "OS resemble a modified cilium". The OS is a modified cilium

4) Intro: "By comparison, the retina of the zebrafish is predominantly cone-based, with approximately 92% cones at the larval stage and about 60% in adults, closely resembling the cone-rich fovea found in humans (Fadool, 2003, Masek, Zang et al., 2023). There is a missing reference here.

5) More clarification about ages could be used in the text and on the figures. What is the range of ages described as "adult"?

6) For Figure 2, need lower magnification panels to show position/organization of the cells within the retina. Panel E needs a scale bar.

7) The relationship between width and length of OS doesn't seem to hold for the other species. Are the scales accurate, since the cell size looks massively different between cone subtypes in some of the fish. Please comment on the length-width ratio in the discussion.

8) Discussion is generally underwhelming, and the comparison to Hebbian plasticity is weak and doesn't really add anything to the paper. Neural activity of photoreceptors is not activated by light (the opposite), which should be made clear. And the LTP comparison is nebulous.

Reviewer #3: The manuscript by Xu et al. reports the intriguing finding that the morphology (length and width) of the outer segments (OS), the light-sensitive region of photoreceptors, is regulated by the kind of the opsin expressed in the OS and the intensity and wavelength of incident light. The authors used elegant genetics in the zebrafish model to establish this, and also present data that argue for a conservation of this principle in other fish species as well as mammals. Overall, the experiments are well designed, the data are nicely presented, and the manuscript is written in a very accessible style.

Despite the novelty of their findings, the shortcoming of the paper is that the authors do not present any data or much of an argument on how photoreceptor-specific opsins could be regulating OS morphology at the mechanistic level. Since the OS is essentially an exaggerated elaboration of the connecting cilia membrane, could it be that the opsins, which are OS membrane proteins, regulate membrane biogenesis and organization? For instance, the opsins could act through the effectors of ciliary membrane biogenesis like the Rab proteins and Arl13b? While this manuscript is suitable for PLoS Biology as a short article and I am not expecting the authors to now put together some mechanistic data along these lines in the revision, it would nevertheless be worthwhile to include some speculation on the mechanism in the discussion section.

Reviewer #4 (Michel Cayouette and Michael Housset signed their report): In this manuscript, Xu et al. address how cells adopt different morphologies. To explore this very interesting and poorly studied question, they use primarily the zebrafish photoreceptor cells as a model system, taking advantage of the various morphologies displayed by photoreceptor subtypes and ease of genetic manipulations in zebrafish. Specifically, they examine how opsins of different spectral properties and ambient light intensity influences photoreceptor outer segment (OS) morphology. They report a positive correlation between OS length and the peak absorption wavelength of the expressed opsin, with an inverse correlation between OS base width and absorption peak. This relationship appears conserved across vertebrate species, including different fish species and rabbit. The authors also show that sws1 mutant zebrafish exhibit UV cone degeneration, which is rescued by ectopic expression of sws1 (UV), mws3 (green), or lws1 (red) opsins. Strikingly, the OS morphology of rescued cones varies depending on which opsin subtype is expressed, supporting a role for opsin sensitivity in shaping photoreceptor morphology. They also show that thyroid hormone (TH) treatment, which shifts opsin expression in red cones to a longer wavelength red opsin, further elongates the OS, reinforcing the link between spectral tuning and OS morphology. The authors propose a mode in which OS geometry is an adaptive response to differences in photon energy and opsin activation efficiency, supported by experiments in dim light and comparisons with deep-sea Sebastes schlegelli.

Overall, this is a very interesting paper addressing a key question in neurobiology. The conclusions are largely supported by the results (although see below), the paper is well written, and the figures are clear and generally convincing. This work will be of interest to vision scientists and to the neuroscience community in general. While the study lacks mechanistic insights into how exactly different opsins leads to changes in cell morphology, these results establish a solid base for future investigation in the field and constitute groundbreaking work in this area. There are, however, a few points that should be clarified and some additional experiments that may help better support the conclusions. Specifically, mechanistic clarity is needed regarding the roles of light versus opsin identity, and additional quantitative and optical analyses are needed to strengthen the central claims. These points are detailed below.

Major comments

1) In fish, as well as chicken and xenopus, photoreceptors dynamically elongate or contract in response to light and dark adaptation, but the implication of this phenomenon on the interpretation of the data presented in this paper is not discussed. It is crucial that the authors discuss their findings in the context of retinomotor movements. This background would help distinguish between reversible, transient changes and more stable OS modifications.

2) While the study convincingly links opsin identity to OS morphology, the interpretation that OS length reflects an adaptation to help increase opsin content is not fully substantiated. OS length and width are not an adequate proxy for opsin quantity because the amount of opsin proteins that can accumulate in OS disk membrane depends on volume. A quantification of OS volume and surface area, ideally supported by 3D reconstructions, would provide a more robust test of the authors' hypothesis of a gain in opsin content in the OS.

3) The manuscript would benefit from additional data to strengthen the model. Specifically, it remains unclear whether OS shape is intrinsically determined by the opsin protein itself or requires downstream effects of light activation. These aspects could be experimentally disentangled by raising fish in complete darkness and under monochromatic light of different wavelength. Moreover, the intriguing possibility that opsin identity dictates OS width while photon absorption regulates OS length deserves more extensive exploration. The authors could for example put transgenics Tg (sw1 :Mws3) and Tg (sws1 :Sws1) of the sws1 mutant background in total darkness until analysis. If OS shape is directly influenced by the light-mediated activity, as predicted by the authors' model, there should not be any difference between the two conditions, since there is no light to activate the opsins.

4) The analogy between OS elongation and dendritic arbor refinement is compelling but unclear. The authors should clarify whether dim-light-induced OS elongation occurs during a critical developmental window, as often observed in dendrite refinement, or can be modified post-developmentally by performing experiments in adult animals. If it is not possible to carry out these experiments, this limitation should at least be discussed.

5) In Figure 1:

* Indicate OS location in panel A.

* Improve contrast for the legend in panel F.

* Specify whether OS width is measured at the base or along the full segment.

6) In Figure 2:

* Panel B: Clarify if the region targeted by the probe lies within, upsteam or downstream of the deleted portion of the sws1 gene.

* Correct the legend: "10 µm in (C-E)" → "10 µm in (C-I)."

* The authors discuss protein levels, but only RNA is quantified in Supplementary Figure 5. Western blotting would be important to confirm protein expression. Also, clarify which genes are being quantified in the figure S5 panel B. Please segregate qPCR done for Sws1 gene and qPCR for Lws1 on different graphics.

7) In Figure 3:

* Change Pseudemys scripta elegans for the correct current taxonomy: Trachemys scripta elegans.

* Panel B: Quantification of OS shape (length and width) is missing for the turtle LWS rescue.

* Panel C: Consider replacing the current schematic with one showing that TH induce expression of longer wavelength red opsin Lws1 at the expense of the shorter Lws2 at the genomic locus.

* Panels F-I: The lipid droplet experiment is technically impressive but lacks clear optical implications. The droplet, likely composed of neutral lipids and localized basally in the myoid, is unlikely to absorb or scatter UV light significantly, contrary to the avian lipid droplet found in birds PRs which contains specific lipids and pigments. Without optical modeling or empirical measurement, the impact on photon flow remains speculative. Conclusions should therefore be toned down and limitations of this approach discussed.

8) In Figure 4:

* Quantify OS volume and base width in addition to length (or change all quantification to volume measurements, as suggested in point 2 above).

* Clarify how the OS tip was defined in overlapping WGA-labeled double cones.

* Address potential morphological artifacts due to delayed fixation in wild-caught specimens; consider using a control feature (e.g., nuclear diameter or IS length) for normalization.

9) In Figure 5: The model presented in not clear. The authors are encouraged to provide a refined, more self-explanatory figure. The position of cones within the retinal layers could be integrated to the model as well as potential effects of wavelength-dependent penetration in the tissue. If additional data can be obtained to distinguish between an intrinsic role of light and opsin protein species in OS width and length, as suggested in point 3 above, this should be considered in the model.

Minor comments

* Regarding the failure to rescue sws1 mutants with tagged opsins, please include a schematic of tag placement and fusion constructs. The C-terminal domain of rhodopsin contains essential trafficking signals, particularly the final 15 amino acids. If this is disrupted by the tag, mislocalization of opsin could explain the lack of rescue. This limitation should be discussed or excluded experimentally.

* In Figure S6H, expression of long-wavelength opsins appears to reduce inner segment (IS) length. Please mention impact of opsin on IS length and discuss whether this reflects coordinated regulation of IS and OS dimensions. Also, the limited rescue by Lws1 compared to rhodopsin raises concerns about expression efficiency or structural compatibility. please clarify what happens to non-rescued rods.

* Clearly state "n" values (number of cones vs. number of animals) for each experiment.

* Indicate developmental stage and retinal region for all quantifications.

* In several figures, the authors used a student's t-test for comparison between multiple conditions, which is not appropriate statistics. ANOVA with post hoc corrections must instead be used.

---

## [Decision Letter · Decision Letter 2]

23 Jan 2026

Dear Dr Zhao,

Thank you for your patience while we considered your revised manuscript "Light intensity and opsin sensitivity shape the morphology of cone photoreceptor outer segments" for publication as a Short Report at PLOS Biology. This revised version of your manuscript has been evaluated by the PLOS Biology editors, the Academic Editor, and three of the original reviewers.

Based on the reviews, we are likely to accept this manuscript for publication, provided you satisfactorily address the remaining points raised by the reviewers. Please also make sure to address the following data and other policy-related requests.

IMPORTANT: Please ensure that your next revision addresses the following editorial and format-related points:

---------------

**Number of figures:

-- As your study was considered as a Short Report, the maximum number of main text figures is 4. Please move elements into the supplement or combine figures to accomplish this -- as discussed separately, we suggest that you make Figure 5 into a Supplementary Figure. Please change references to the figure accordingly.

**Financial disclosure statement:

-- Please include in your Financial disclosure statement the answer to the following question: "Did the sponsors or funders play any role in the study design, data collection and analysis, decision to publish, or preparation of the manuscript?".

-- Please also add links to the funding agencies.

**Ethics:

-- Please include the specific national or international regulations/guidelines to which your animal care and use protocol adhered. Please note that institutional or accreditation organization guidelines (such as AAALAC) do not meet this requirement.

**Data:

-- Thank you for including the underlying data in Table S3. We require a few changes to this:

-- Please change the title of this item to "S1 Data" (upload as "S1_Data.xlsx") and refer to it this way throughout the paper.

-- In the supplementary data file, we think that the panel description for the S8 tab is incorrect. Please fix.

-- Please add supplementary data for the figure currently found in the movie file (also see below as this figure will also need to be moved and given a title and description).

-- Please cite the location of the data clearly in all relevant main and supplementary Figure legends, e.g. “The data underlying this Figure can be found in S1 Data” or “The data underlying this Figure can be found in https://doi.org/10.5281/zenodo.XXXXX”

**Supplement:

-- Please move the supplementary figure legends, and the supplementary tables into the main manuscript document. Please upload all supplementary figures separately as Supporting Information files.

-- We noticed that your supplementary movie file contains a figure. Please give this figure its own name (Figure S9, for example) and include a legend, and treat it the same as the other supplementary figures. Please title the supplementary movie file as "S1 Movie" and refer to it as such throughout the paper.

**Code availability:

Per journal policy, if you have generated any custom code during the course of this investigation, we require that you make it available without restrictions. Please ensure that the code is sufficiently well documented and reusable, and that your Data Statement in the Editorial Manager submission system accurately describes where your code can be found.

---------------

We expect to receive your revised manuscript within two weeks.

*Published Peer Review History*

*Press*

Sincerely,

Taylor

Taylor Hart, PhD,

Associate Editor

thart@plos.org

PLOS Biology

Reviewer remarks:

Reviewer #1 [Juan Angueyra]: The authors addressed our comments satisfactorily, including reassessing parallels with Hebbian plasticity, improving description of methods and structural rearrangements to improve readability. The concerns of other reviewers have also been sufficiently addressed. We believe these edits have improved the paper and strengthened their models to explain the mechanisms that may dictate the structure of photoreceptor outer segments in relation to the environment. We only have two minor suggestions for revisions of the specific interpretations that won't require any additional experiments, and that we leave to the authors discretion.

First, the following paragraph is still inaccurate and/or confusing:

"Light energy acquisition depends on two main factors: the energy of individual photons and the number of photons captured by photopigments. Reducing photon energy (e.g., using longer wavelengths) or limiting photon capture (e.g., through additional lipid droplets or under dim-light conditions) lowers the probability of opsin activation".

As we previously stated, just using longer wavelengths does not guarantee less opsin activation, which this sentence seems to imply. If the authors intended "to emphasize that, for a given opsin at its optimal wavelength, higher-energy photons generally have a higher probability of causing isomerization than lower-energy photons at their respective optimal wavelengths", this should be incorporated in the manuscript, although we still find even this explanation confusing. Since the "number of photons captured by photopigments" depends on photon energy/wavelength AND spectral sensitivity of the photopigment, this phrasing hides an important logical step. I would argue that it's clearer to say that photon capture depends on two main factors: the total number of photons and the relation between the spectral composition of these photons and the spectral sensitivity of the photopigment. Lowering opsin activation requires decreasing the number of photons that reach the opsins or decreasing the spectral overlap between the stimulus and the opsin.

Second, it might be worth elaborating or clarifying a bit more on what the authors consider as the most likely biological implementation of how "neural activity" leads to changes in OS length. As briefly mentioned, transducin activation and glutamate release in photoreceptors are inversely related, so that "when fish are raised in darkness", there is indeed minimal transducin activation but glutamate release by photoreceptors is at its highest, which could be considered maximal neural activity? Do the authors assume that this could be processed through circuits in the ON pathway, which flips the polarity of signals? Is that more likely than a cell-autonomous mechanism depending directly on the number of activated transducins in individual cones? Disclosing glutamate release in the final model figure may make this explicit for readers.

Reviewer #2: The revised paper is strong and acceptable for publication, with a couple of suggested minor changes.

1. The scale bar in Fig4 labelled as "10 cm in (D)" should be "10 cm in (I)".

2. The WGA staining in the manuscript is clearly labelling cones. However, this is unusual. I'm unclear if that's a result of a different WGA preparation, more concentrated solution, or imaging parameters. It should be mentioned in the manuscript that WGA is typically used a rod OS marker, but that the researchers found it could be adapted for cone labelling. Otherwise, it will confuse the reader.

3. Also, it appears in many of the panels that WGA fails to label the top portion of the UV cone OS. If so, this should be mentioned.

4. The panels and labels in Fig S6 are very small. The figure could be reorganized to enlarge panels.

Reviewer #4 [Michel Cayouette and Michael Housset]: The authors have done a wonderful job addressing our comments both with text changes and new experiments. We have no further comments. Congratulations on a very interesting study.

---

## [Editor Report · Decision Letter 3]

30 Jan 2026

Dear Dr Zhao,

Thank you for the submission of your revised Short Report "Light intensity and opsin sensitivity shape the morphology of cone photoreceptor outer segments" for publication in PLOS Biology. On behalf of my colleagues and the Academic Editor, Tom Baden, I am pleased to say that we can in principle accept your manuscript for publication, provided you address any remaining formatting and reporting issues. These will be detailed in an email you should receive within 2-3 business days from our colleagues in the journal operations team; no action is required from you until then. Please note that we will not be able to formally accept your manuscript and schedule it for publication until you have completed any requested changes.

PRESS

Sincerely,

Taylor

Taylor Hart, PhD,

Associate Editor

PLOS Biology

thart@plos.org